# Phenomenological model of suspended sediment transport in a small catchment

**Amande Roque-Bernard**[1], **Antoine Lucas**[1], **Eric Gayer**[1], **Pascal Allemand**[2], **Céline Dessert**[1], **and Eric Lajeunesse**[1]

[1]Université Paris Cité, Institut de physique du globe de Paris, CNRS, F-75005, Paris, France
[2]Université de Lyon, Université Lyon 1 & ENS Lyon & CNRS, Laboratoire de Géologie de Lyon, Terre Planètes Environnement, UMR 5276, 69100 Villeurbanne, France

**Correspondence:** A. Roque-Bernard (a.roqber@gmail.com), A. Lucas (lucas@ipgp.fr)

**Abstract.** We develop a phenomenological model of suspended-sediment transport on the basis of data acquired in the Capesterre river, which drains a small tropical catchment in Guadeloupe. The model correctly represents the concentration of suspended sediment during floods, provided that the relation between concentration and water-level forms a counterclockwise loop. In the model, the properties of the sediment and of the river are all lumped into four parameters : a settling velocity related to the size of the suspended sediment, a threshold water-level which acts as a proxy for the threshold shear stress, a characteristic entrainment rate, and a dimensionless exponent. The value of the parameters change from one flood to the next, probably reflecting changes in the characteristics of the river and the fine sediment. Finally, a test of the model against data acquired in a small catchment in the french Alps, suggests that the model is versatile enough to be used in diverse hydrological settings.

## 1 Introduction

Rivers transport sediments according to their particle size. Coarse particles, like gravels or pebbles, bounce and roll above the sediment bed (Bagnold, 1973). Finer sediments, like silt, clay or even sand behave differently: their falling velocity is comparable to the fluctuations of the flow velocity induced by turbulence, and they remain suspended in the water column, trapped by turbulent eddies (Van Rijn, 1984). They thus advance with the flow, until they eventually settle back on the river bed (Phillips et al., 2019). This suspended load is often the main contribution to the sediment discharge that a river carries out of its watershed (Turowski et al., 2010; Liu et al., 2011). It is therefore a major component of the erosion of continental surfaces (Summerfield and Hulton, 1994; Syvitski et al., 2003). Yet the matter suspended in a river does not entirely consists of sediment; it also includes nutrients, particulate organic matter, carbon, micro-plastics, colloidal particles, and various contaminants (e.g., D'Avignon et al., 2022). Suspended load thus affects the quality of water and riverine ecological habitats (Suttle et al., 2004; Battin et al.,

2008; Lloret et al., 2013; Koiter et al., 2013). It is therefore critical to assess this suspended load.

The simplest way to estimate the suspended load carried by a river is to filter a sample of water collected in the stream, at a point where water is well mixed. The weight of the filter, once it is dried, gives a measure of the concentration of sediment, which, combined with the flow discharge, yields the rate of suspended sediment transport (Bierman and Montgomery, 2014). In practice, however, this long and tedious procedure is inappropriate for high frequency measurements. Instead, in-situ monitoring of the suspended load often relies on the use of a turbidimeter, an instrument capable of measuring the turbidity of the river at a high frequency and over long periods of time (Turowski et al., 2010; Esteves et al., 2019). Turbidity measures the amount of light scattered by the suspended particles in the water column. It is thus a convenient proxy for the suspended load concentration. Turbidity, however, also depends on the size distribution of the suspended particles, on their shape, and on their chemical composition. Its conversion into a sediment concentration thus requires an on-site calibration, which ultimately relies on the filtering of water samples (Minella et al., 2008).

Field measurements show that the concentration of suspended sediment fluctuates with the river discharge, and culminates during floods. Based on this observation, it is tempting to use discharge as a proxy for suspended load (e.g., Ahn and Steinschneider, 2018). Yet, the relation between discharge and concentration is not univocal: when observed at the scale of a single flood event, it often exhibits a hysteretic loop (e.g., Williams, 1989). These loops are observed under various geological and climatic settings, independently of the size of the catchment that the river drains (Langlois et al., 2005; Bača, 2008; Eder et al., 2010; Ziegler et al., 2014).

Many factors combine to shape this hysteretic behavior. First of all, the suspended load concentration adjusts to the local shear stress over a characteristic time that depends on the flow depth and the particle settling velocity (Claudin et al., 2011). The resulting delay between discharge and sediment concentration induces a counterclockwise loop. In gravel bed rivers, fine particles are often trapped below a layer of coarse sediments such as pebbles, a phenomenon known as armoring (Frey and Church, 2009; Ferdowsi et al., 2017a). The river bed then acts as a sediment buffer that stores and releases fine particles according to its own dynamics, altering the shape of the concentration-discharge relationship (Orwin and Smart, 2004; Turowski et al., 2010; Park and Hunt, 2017; Guillon et al., 2018; Misset et al., 2019b). Fine particles, however, do not always originate from the river bed. During storms, hillslope runoff, landsliding, and bank erosion may also feed the river with a significant amount of fine particles (Hovius et al., 2000; Allemand et al., 2014). In this case, the spatial distribution of rainfall in the catchment area and the distance between the sediment sources and the sampling point influence the shape of the concentration-discharge relationship in a complex way (e.g., Asselman, 1999; Smith and Dragovich, 2009; Misset et al., 2019a). Finally, if the velocity of the flow that carries the suspended sediment is smaller than the celerity of the flood wave, the resulting delay between the discharge and the sediment peak induces a counterclockwise loop. This effect may dominate the discharge-concentration relationship if the distance traveled by the suspended particles is long enough (Klein, 1984; Nistor and Church, 2005).

The variety of processes involved in the transport of suspended sediment has led to the development of different types of model (Gao, 2008; Vercruysse and Grabowski, 2019). Following the pioneering work of Rouse (1939), a large body of work focused on the formulation of mathematical models that explain how the balance between turbulent diffusion and sedimentation determines the vertical profile of the suspended-sediment concentration in a turbulent flow (Van Rijn, 1984; Garcia and Parker, 1991; Wright and Parker, 2004; Claudin et al., 2011). Combined with equations for the flow, these so called "hydromorphodynamic models" provide a satisfactory physical picture of the transport of suspended sediment in rivers (Van Rijn, 2007; Bouchez et al., 2010; Armijos et al., 2017). However, their use to predict sediment

transport requires the calibration of a large number of hydrological parameters, as well as a precise knowledge of the river topography and discharge (Lepesqueur et al., 2019).

Hydrological models do not incorporate the input of fine sediment from the hillslopes surrounding the river. Yet, in a small catchment, this contribution may represent a significant fraction of the total sediment yield during rainfall (Gao, 2008). The need to account for hillslopes processes motivated the development of "distributed models". Distributed models break down the catchment into several regions. In each of them, the susceptibility to rainfall-induced soil erosion is parameterized by a series of indices that describe local properties such as soil characteristics, land cover, land use, and topography (Wischmeier and Smith, 1978; Renard et al., 1991, 2017). Based on the value of these indices, the model solves a system of equations that account for various hillslopes and hydrological processes, and predicts the sediment yield at the catchment outlet (De Aragão et al., 2005). These models explicitly account for the location of the sediment sources inside the catchment. However, they involve a large number of adjustable parameters, at the risk of overfitting the data (White, 2006).

Recently, the increase in the volume of environmental data has led to the development of methods based on data mining and artificial intelligence (Vercruysse and Grabowski, 2019). The principle is to use multivariate data-analysis to identify the variables that primarily control the concentration of suspended sediment, and reconstruct the function that relates them together. This approach – which relies on methods ranging from quantile regression forests to artificial neural networks – successfully predicts the concentration of suspended sediment in various contexts (Cobaner et al., 2009; Bilotta et al., 2012; Zimmermann et al., 2012). In particular, it allows to identify correlations between complex hysteresis patterns and antecedent hydro-meteorological conditions (Perks et al., 2015). This method, however, is essentially a " black box" approach, and prediction often comes at the cost of understanding (Vercruysse and Grabowski, 2019). Besides, it requires the acquisition of large datasets over multiple timescales, which restricts its application to a few well instrumented catchments.

Despite this impressive body of work, modeling and predicting the entrainment, propagation, and deposition of suspended particles is still a challenging task. Here, we use data acquired in the Capesterre river, a gravel bed river that drains a small watershed on the south-east coast of Basse-Terre Island (Guadeloupe archipelago), to develop a phenomenological model of suspended-sediment transport. We begin with a description of the Capesterre catchment, which is equipped with a gauging station that records high-frequency measurements of the river discharge and turbidity (section 2). Based on the data collected there, we develop a phenomenological model of suspended sediment transport, that accounts explicitly for the exchange of small particles between the river bed and the water column (section 3). We then test this model

against the data collected in Capesterre, and find that it correctly represents the concentration of suspended sediment during floods, provided that the relation between concentration and water-level forms a counterclockwise loop (section 4). Encouraged by this result, we discuss the physical meaning of the model parameters, and demonstrate its versatility by applying it to a second catchment, the Draix-Laval catchment in the french Alps (section 5). Finally, we discuss the strengths and limitations of the model (section 5) before concluding.

## 2 The Capesterre catchment

### 2.1 Field site and measurements

We begin our investigation in the Capesterre river, located on Basse-Terre Island. This volcanic island of the Guadeloupe archipelago belongs to the subduction arc of the Lesser Antilles (Feuillet et al., 2002) (Fig.1a). Basse-Terre climate is tropical : daily temperatures range between 24°C and 28°C, and the annual rainfall rate is about 5200 mm y$^{-1}$. The combination of this tropical climate with a steep volcanic relief produces high erosion rates, that range between 800 and 4000 t km$^{-2}$ y$^{-1}$ (Rad et al., 2006; Dessert et al., 2015). These values place Basse-Terre Island among the fastest eroding spots on Earth (Summerfield and Hulton, 1994).

On Basse-Terre, rainfalls are intermittent and occur mainly as short, high-magnitude events. As a result, the discharge of rivers varies abruptly, with frequent flash floods triggered by tropical rainfalls and hurricanes (Fig.1b and c). These extreme climatic events, particularly frequent during the rainy season from June to January, trigger landslides, rock avalanches and debris flows, which are the main drivers of erosion (Allemand et al., 2014).

The Capesterre river drains a steep watershed, on the windward side of the active Soufrière volcano (Fig 1a). The mean annual rainfall rate is 5700 mm y$^{-1}$. However, topography induces a strong orographic gradient, and the intensity of rainfalls is highly heterogeneous within the catchment.

Capesterre catchment is underlain by an andesitic bedrock, aged from 400 to 600 ky (Samper et al., 2007). Soils mostly consist of thin andosols of thickness typically less than 1 m (Colmet-Daage and Bernard, 1979; Lloret et al., 2016). The Capesterre river flows over 19.7 km, from its headwater on the flanks of the active volcano, at an altitude of 1390 m asl, down to the Capesterre village, where it discharges into the Atlantic ocean. Its channel is made of bedrock, partly covered by a thin layer of alluvial sediment. Three kilometers from the sea, the river suddenly turns alluvial, as its slope gradually decreases.

The "Observatoire de l'Eau et de l'éRosion aux Antilles" (ObsERA) operates a gauging station at the site of "La Digue", at an altitude of 189 m asl, a few hundred meters upstream of the point where the river turns alluvial (Fig.1b and c). At the station, the river drains a catchment of area 16.4 km$^2$, almost entirely located within the boundaries of the National Park of Guadeloupe. There, a thick rainforest limits the input of sediment from hillslopes, and anthropogenic forcing is weak. Based on a 4-years water-sampling campaign, Lloret et al. (2013) estimated that the mean flux of suspended matter is about 153 t km$^{-2}$yr$^{-1}$. About 10% of this flux consists of Particulate Organic Carbon.

La Digue's station is equipped with a pressure sensor (CS451, Campbell Scientific Inc.) which measures the river stage relative to the fixed datum defined by a staff gauge installed by the "Direction de l'Environnement, de l'Aménagement et du Logement" (DEAL-Guadeloupe). In addition, a turbidimeter (OBS3+, Campbell Scientific Inc.) measures the turbidity of the water. Both pressure and turbidity sensors are connected to a data-logger (Campbell CR800), which records their respective measurements every 5 minutes since 2013 (Fig.1e and f). The station is also equipped with an automatic water sampler (6712 Full-Size Portable Sampler) triggered by a pressure probe (ISCO 720 Submerged Probe module). This device collects water samples when the river stage exceeds a threshold set by the operator (about 50 cm). Filtration of these samples allows us to measure the concentration of suspended material $C$ (expressed in mg L$^{-1}$), and to calibrate the relation between the latter and the turbidity $T$ (NTU) (Fig.1d). A fit of a power law through our data yields $C = \alpha T^n$, with $\alpha = 0.26 \pm 0.04$ mgL$^{-1}$ and $n = 1.23 \pm 0.04$.

The data acquired in the Capesterre river show that the water stage fluctuates, as floods follow each others (Fig.1e). Each flood starts with an abrupt increase of the water stage — the latter may rise from 0.1 to 1m in less than 1 hour — followed by a slow recession which last for 12 to 20 hours. In between two floods, the water stage remains low (i.e. 10 to 20 centimeters). The concentration of suspended material, calculated from turbidity, is highly intermittent (Fig. 1f) : it is virtually zero 98% of the time, and rises only during floods, where it may reach up to 1430 mg L$^{-1}$.

### 2.2 Concentration of suspended material

Sediment transport in the Capesterre river occurs only during floods. We therefore extract four flood events from our dataset and observe the evolution of the water level and of the concentration of suspended sediment in more details (Fig. 2 and 3). All four floods exhibit the same simple shape : a steep rise, followed by a slow decline. Visual inspection of the data reveals that this pattern is representative of the entire data set, with multi-peak events representing less than a few percent of floods.

During a flood, the concentration of suspended sediment follows the fluctuations of the water stage: it is equal to zero before the flood, rises when the water stage exceeds a threshold of about 20 centimeters, increases with the water stage until the flood peak, and finally decreases back to zero during the recession limb of the flood (Fig. 2, left panels).

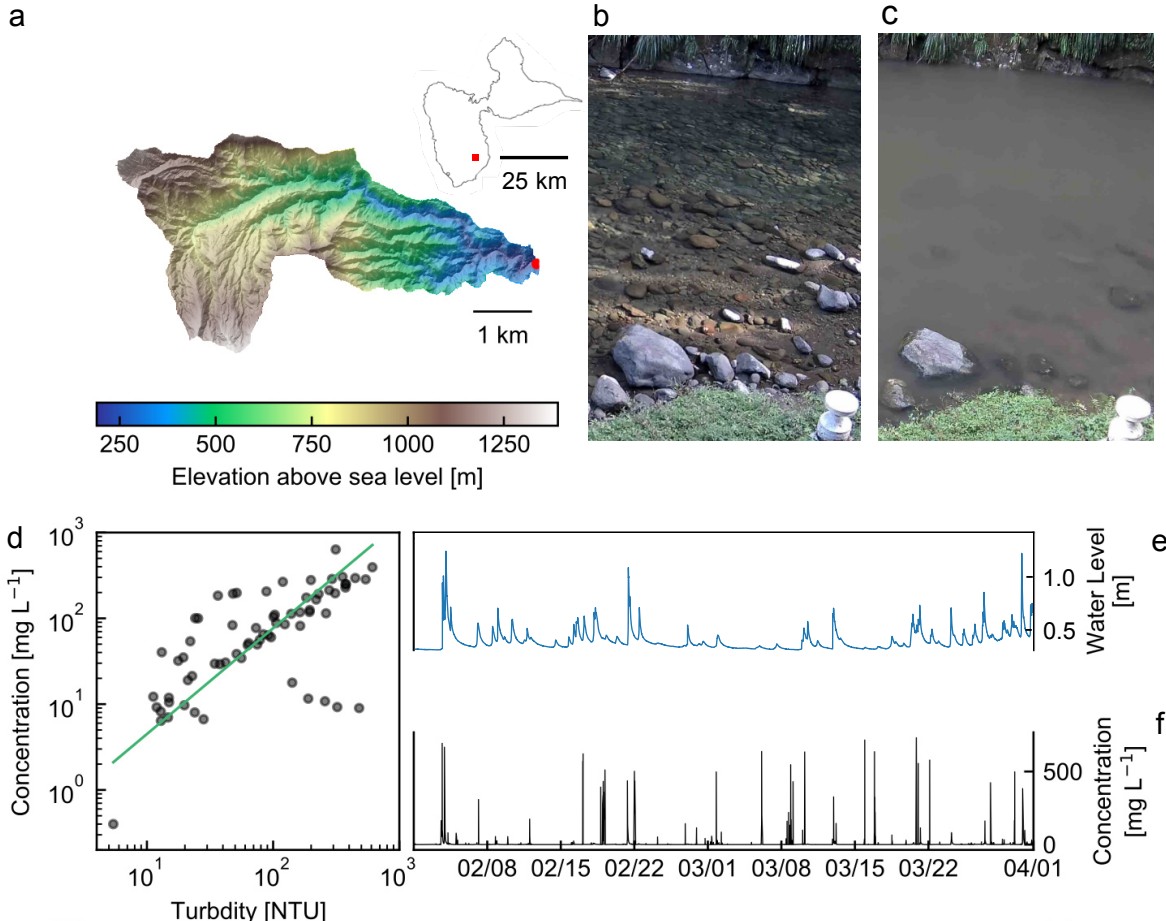

**Figure 1.** Capesterre catchment on Basse-Terre Island, Guadeloupe, French West Indies. (a) Topographic map of the catchment. Inset: map of Guadeloupe (IGN RGE ALTI® 10 m). Red dots locate the catchment outlet on both maps ($16°04'19''$N, $61°36'33''$W). (b) and (c) Images of the river at low and high stage, respectively. (d) Concentration of suspended sediment vs turbidity in the Capesterre river. (e) Water level and (f) concentration of suspended sediment from February 1st, 2021 to April 1st, 2021.

A closer look at the data, however, reveals a time lag between the flood peak and the concentration peak. For instance, for the three floods shown in figure 2, the concentration peak occurs about 30 minutes after the flood peak. A plot of the concentration as a function of the water stage better highlights this time lag between concentration and water-level: the concentration vs water-level relation forms an counterclockwise hysteretic loop (Fig. 2, right panels).

Because we deduce the concentration of suspended sediment from the river turbidity, interpreting the origin of concentration loops requires some caution. Indeed, concentration loops may sometimes be an artifact caused by a change of the turbidity-concentration relationship in the course of a flood (Landers and Sturm, 2013). This happens when the size distribution of the suspended load evolves significantly during a flood. The ObsERA observatory used a LISST-Streamside to measure the size distribution of the particles in suspension in the Capesterre river during a flood in 2010. Their data reveal a bimodal distribution with two peaks (see section 5.1.1). The first one, around 5 micrometers, corresponds to the washload, and remains remarkably stable during the flood. The position of the second peak varies between 23 and and 32 micrometers. Assuming that this flood is representative, we expect no significant change of the turbidity-concentration relationship during a flood. Moreover, direct measurements of the concentration of suspended sediment, based on manual water-sampling, confirm that the concentration vs water-level relation forms hysteretic loops (Lloret, 2010; Lloret et al., 2013). Based on these observations, we therefore attribute the hysteresis of the turbidity of the Capesterre river to an hysteresis of its concentration of suspended sediment.

In the Capesterre river, the flood peak is not systematically ahead of the concentration peak. Sometimes, it is the concentration peak that precedes the flood peak, and the concentration vs water-level relation then forms a clockwise hysteretic loop (Fig. 3).

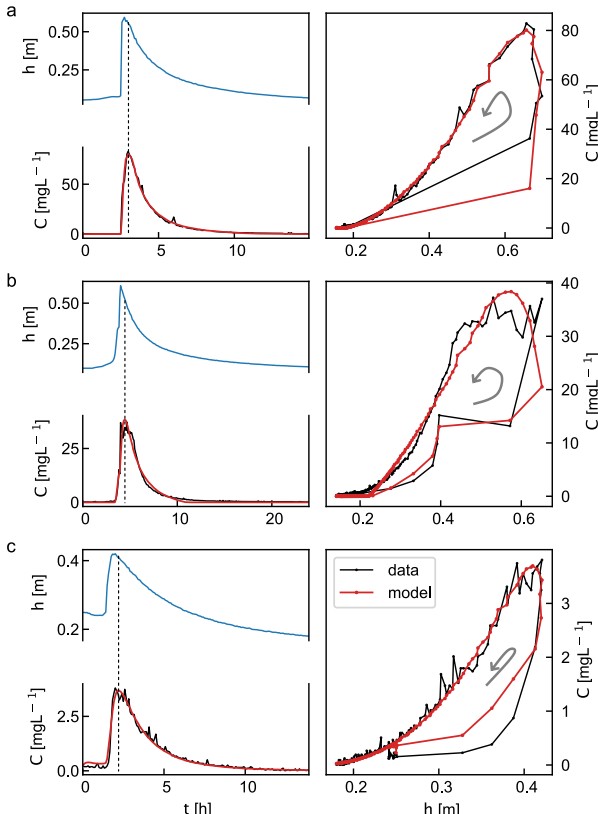

**Figure 2.** Three floods in the Capesterre river for which the concentration vs water-level relation forms a counterclockwise loop. (a) from 05/09/2019, 19:00 to 06/09/2019, 10:00. (b) from 22/07/2019, 22:00 to 23/07/2019, 10:00. (c) from 03/11/2021, 10:00 to 04/11/2021, 00:00. Left panels: time series of the water level (blue line) and the concentration of suspended sediment (black line) measured at the gauging station. The vertical dashed-lines indicate the moments when the concentration reaches its peak value. Right panels : relation between concentration and water-level. Gray arrows indicate the direction of the hysteresis loops. On each panel, the red line is the concentration predicted from the best-fit model.

To determine the proportion of clockwise versus counterclockwise loops in the Capesterre river, we first extract individual flood events from our dataset. To do so, we define a flood as a period during which the water stage is higher than

5 a threshold set to 20 cm. This value roughly corresponds to the threshold above which the flow is strong enough to carry suspended sediment. Based on this definition, we isolate 217 individual floods between May 2019 and December 2021.

We must now determine the direction of the concentration

10 vs water-level relation for each of these floods. Inspired by Langlois et al. (2005) and Misset et al. (2019a), we define, for each flood, the hysteresis index as

$$I_\mathrm{H} = \int \widetilde{C} d\widetilde{h}, \tag{1}$$

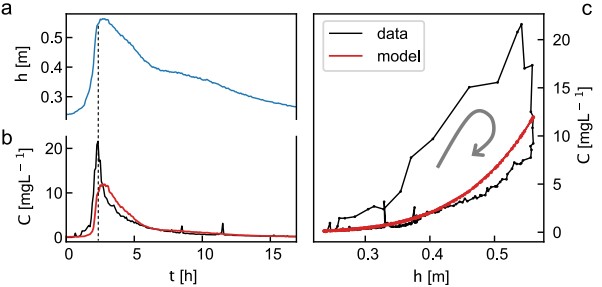

**Figure 3.** Flood recorded in the Capesterre river from 15/01/2021, 7:00 to 16/01/2021 00:00. During this event, the concentration vs water-level relation formed a clockwise loop. (a) time series of the water level (blue line), and (b) the concentration of suspended sediment (black line) measured at the gauging station. (c) relation between concentration and water-level. Gray arrows indicate the direction of the hysteresis loops. The vertical dashed-line indicates the moment when the concentration reaches its peak value. On each panel, the red line is the concentration predicted from the best-fit model.

where we introduce the normalized water level, $\widetilde{h} = (h - h_\mathrm{min})/(h_\mathrm{max} - h_\mathrm{min})$, and the normalized concentration, 15 $\widetilde{C} = (C - C_\mathrm{min})/(C_\mathrm{max} - C_\mathrm{min})$. $h_\mathrm{min}$ and $h_\mathrm{max}$ are the minimum and maximum values of the water level $h$ during the flood. Similarly, $C_\mathrm{min}$ and $C_\mathrm{max}$ are the minimum and maximum values of the concentration $C$ during the flood. With these definitions, both the normalized water-level and the 20 normalized concentration vary between 0 and 1. Accordingly, the hysteresis index $I_\mathrm{H}$ ranges between -1 and 1. Negative values correspond to counterclockwise hysteresis, while positive ones indicate clockwise loops.

To characterize the proportion of clockwise versus coun- 25 terclockwise loops, we just need to compute the hysteresis index $I_\mathrm{H}$ for each of the 217 floods of our catalog. We find that this index ranges between -0.719 and 0.452, with a mean value of -0.030, a median of -0.021 and a standard deviation of 0.178. A plot of the cumulative distribution of 30 the hysteresis index reveals that 61% of the floods detected on Capesterre are characterized by a counterclockwise loop (Fig.4a).

To detect a possible influence of the season, we categorize the flood events by wet (177 events) and dry (44 events) 35 seasons. We then compute the distributions of the hysteresis index for each season, and find that clockwise and counterclockwise hysteresis are equally present in both seasons, with median values very close to zero : -0.007 for the dry season and -0.030 for the wet one (Fig.4b). 40

To summarize, about 61% of the floods of the Capesterre river exhibit a counterclockwise hysteresis loops, with no influence of the season. In the next section, we therefore focus on the counterclockwise hysteretic loops, and formulate a simple model, that accounts for their shape. 45

## 3   Phenomenological model of suspended-sediment transport

In this section, we develop a model to account for the evolution of the concentration of suspended sediment in the Capesterre river. Given the complexity of the problem, our objective is not to establish a comprehensive physical theory of suspended sediment transport, but, rather, to derive a phenomenological equation that reproduces our field measurements. We therefore start with a series of simplifying assumptions, driven by field observations.

We first note that the Capesterre catchment is densely vegetated. Accordingly, we assume that, during a flood, the quantity of fine sediment that hillslopes deliver to the river is negligible compared to that originating from the river bed. We therefore treat the river bed as the sole source of sediment. We furthermore assume that the quantity of fine sediment stored in the river bed is so large that the entrainment of sediment is not limited by supply, a condition often referred to as "transport limited" (Dietrich et al., 2003). Of course, these two assumptions only hold at the scale of a flood event. Over the longer term, hillslopes processes gradually replenish the river with sediments, compensating for the erosion of the bed during floods. The time scale of this process is uncertain, but it is likely much longer than a few weeks. During extreme rainfall events, landslides or rock avalanches as well as debris flows may also feed the river with large amounts of sediment. When this happens, the sediment bed cannot be considered the only source of suspended sediment, and our model might fail to represent its concentration.

In the Capesterre river, the water level rarely exceeds 1 meter, a value comparable to the median grain size of its sediment bed, of the order of ten centimeters. As suspended particles mainly consist of silt and fine sand of size less than 100 $\mu$m (see section 5.1.1), we expect that the turbulence induced by the bed roughness is high enough to homogenize the concentration of suspended sediment in the river, an assumption supported by images of the river during floods (Fig. 1c). Accordingly, we neglect any vertical or lateral gradient of the concentration. We discuss in more details the limits of this assumption in section 5.1.1.

Within the previous set of assumptions, conservation of the mass of suspended sediment reads :

$$\frac{\partial}{\partial t}(WhC) + \frac{\partial}{\partial x}(uWhC) = W(E - D), \qquad (2)$$

where $C$ is the concentration of suspended sediment (mass per unit volume), $h$ is the water-level in the river, $u$ is the average flow velocity, $W$ is the river width, $x$ is the streamwise coordinate along the river course, and $t$ is time. $E$ is the entrainment rate, i.e. the mass of sediment entrained from the river bed per unit time and area. Conversely, $D$ denotes the deposition rate, i.e. the mass of sediment deposited on the river bed per unit time and area.

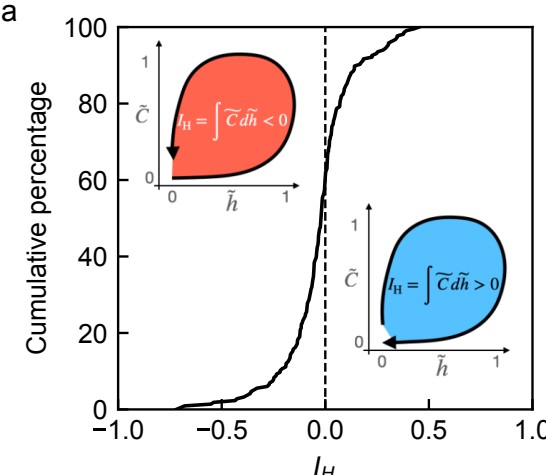

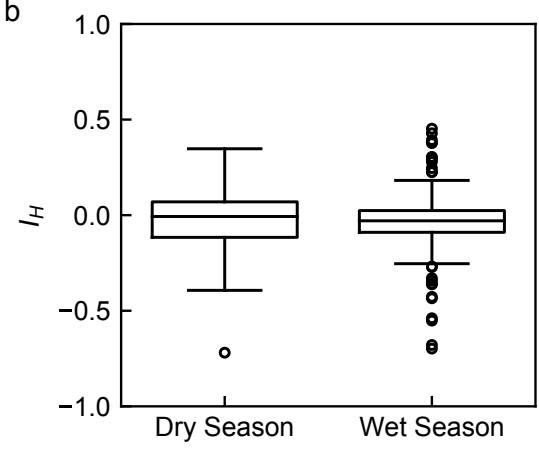

**Figure 4.** (a) Cumulative distribution function (CDF) of the hysteresis index $I_H$ in Capesterre. Insets: cartoons illustrating the physical meaning of the hysteresis index, $I_H$. The magnitude of $I_H$ is equal to the colored area, while its sign depends on the direction of the hysteresis loop : it is positive for a clockwise loop (blue area) and negative for a counterclockwise one (red area). (b) Box plot of the hysteresis indices according to the season. The box extends from $Q_1$ to $Q_3$ of the data. Its central line represents the median ($Q_2$). The whiskers extend from the box by $1.5 \times (Q_3 - Q_1)$. Fliers are outliers.

To compute the concentration of suspended sediment, we need to supplement equation (2) with expressions for the entrainment and deposition rates. Particles in suspension in the river settle under the action of gravity. In the Capesterre river, the concentration of particles never exceeds $0.9 \text{ g L}^{-1}$, a value small enough to neglect interactions between particles. In this dilute regime, we expect that the settling velocity of a particle is independent of the concentration. It depends, however, on the particle size. The suspended load usually consists of grains of different sizes, which thus settle with different velocities. However, following Van Rijn (1986), we further simplify our model, and assume that the decline in total sediment concentration can be represented with a single representative settling velocity, $V_s$. We shall discuss in more details the limits of this assumption in section 5.1.1. For the moment, we use it to derive the deposition rate which reads:

$$D = C V_s. \qquad (3)$$

Let us now turn our attention to the entrainment rate. The river entrains sediment when the shear stress it exerts on its bed exceeds a threshold value (Shields, 1936). Calculating this shear stress requires additional equations, which also means additional assumptions and parameters. To avoid this inconvenience and keep the model as simple as possible, we note that an increase of water level often implies an increase of bottom shear stress. In the following, we shall therefore use water level as a proxy for shear stress. Indeed, field data show that the Capesterre river carries sediment only when the water level exceeds a threshold of approximately $h_t \approx 20\text{cm}$. Accordingly, we look for an expression of the entrainment rate in the form $E = f(h - h_t)$, where $f$ is some unknown function. For lack of additional constraint, we choose the simplest possible form, and propose that:

$$E = \epsilon \left( \frac{h}{h_t} - 1 \right)^n H \left( \frac{h}{h_t} - 1 \right), \qquad (4)$$

where $\epsilon$ is a characteristic entrainment rate (mass per unit time and bed area), $n$ is a dimensionless exponent, and $H$ is the Heaviside function. Keeping in mind that the water level $h$ acts as a proxy for the shear stress, equation (4) is similar to expressions of the entrainment rate commonly used in the literature (Bagnold, 1956; Van Rijn, 1986, 2007; Claudin et al., 2011).

Combining the expression of the deposition (3) and erosion (4) rates with the mass balance (2) yields the following equation:

$$\frac{\partial(hC)}{\partial t} = \left[ \epsilon \left( \frac{h}{h_t} - 1 \right)^n H \left( \frac{h}{h_t} - 1 \right) - C V_s \right] - \frac{1}{W} \frac{\partial(WuhC)}{\partial x}, \qquad (5)$$

where we assume that the river width does not change significantly over the duration of a flood. The evolution of the concentration of suspended sediment (Eq. (5), left term) is thus controlled by the balance between two terms. The first one (Eq. (5), right member, first term) is a local term which accounts for the difference between particle entrainment and deposition. The second one (Eq. (5), right member, second term) is an advective term, which accounts for the streamwise variations of the flux of suspended sediment, $F = WuhC$.

Streamwise variations of the suspended-sediment flux can result from an inhomogeneous distribution of sediment sources along the river course. We have, however, explicitly excluded this possibility from the model assumptions. The other possible cause of streamwise variations is the formation of a flood wave : during rainfall, groundwater and overland flow discharge water into the river, and generate a surge that propagates downstream (Alsdorf et al., 2005; Guérin et al., 2019). This flood wave induces variations of the flow depth and velocity, and therefore of the sediment flux in the streamwise direction, whose effects are described by the spatial derivative of the advective term (Lepesqueur et al., 2019).

Before trying to solve equation (5), we perform a scaling analysis to estimate the magnitude of each of its terms (Barenblatt and Isaakovich, 1996). First, we note that equation (5) involves 3 parameters : the threshold water level $h_t$, the particle settling-velocity $V_s$, and the characteristic entrainement rate, $\epsilon$. Accordingly, we introduce the rescaled water level $\tilde{h} = h/h_t$, the rescaled concentration, $\tilde{C} = (V_s/\epsilon)\, C$, and the rescaled time, $\tilde{t} = t/\tau_s$, where we define the characteristic settling time $\tau_s = h_t/V_s$. The latter corresponds to the time necessary for a particle to settle at the representative settling velocity $V_s$ over a height equal to the threshold height, $h_t$. To turn equation (5) into a dimensionless equation, we still need a characteristic scale for the streamwise coordinate, $x$. The $x$ derivative in equation (5) corresponds to the streamwise gradient of flux induced by the formation of a flood wave. We thus expect $x$ to scale like the characteristic length of the flood wave, $\ell \sim U\tau_f$, where $\tau_f$ is the duration of the flood and $U$ is the characteristic flow velocity. We therefore define the rescaled streamwise coordinate as $\tilde{x} = x/(U\tau_f)$, and the rescaled flow velocity as $\tilde{u} = u/U$. With these new variables, we write the conservation of mass (5) in its dimensionless form :

$$\frac{\partial(\tilde{h}\tilde{C})}{\partial \tilde{t}} = \left( \tilde{h} - 1 \right)^n H \left( \tilde{h} - 1 \right) - \tilde{C} - \frac{\tau_s}{\tau_f} \left[ \frac{1}{W} \frac{\partial}{\partial \tilde{x}} (W\tilde{u}\tilde{h}\tilde{C}) \right]. \qquad (6)$$

Scaling analysis thus reveals that the relative magnitude of the advective term depends on the ratio between the characteristic particle settling time and the duration of the flood. When the flood lasts much longer than the settling time, the advective term can be neglected: the flood wave is so large that the streamwise gradient of the sediment flux becomes negligible. The evolution of the concentration of suspended material is then controlled by the local balance between the entrainment and the deposition rates. In the Capesterre river,

field measurements indicate that : (1) the threshold for grains entrainment is approximately $h_t \approx 20$ cm, (2) the median grain size of the suspended load is about 40 $\mu m$ (see section 5.1.1). Assuming that the density of the particles is $\rho_s$ = 2700 kg m$^{-3}$, this corresponds to a settling velocity of the order of $V_s \sim 1.5 \cdot 10^{-3}$ m s$^{-1}$. With these values, the characteristic settling time, $\tau_s \approx 130$ s, is very short compared to the typical duration of a flood, $\tau_f \gtrsim 12$ hours, so that $\tau_s/\tau_f \lesssim 3 \cdot 10^{-3}$. Therefore, we neglect the advective term in the following.

With this approximation, and coming back to dimensional variables, the conservation of mass becomes :

$$\frac{d\phi}{dt} = \epsilon \left( \frac{h}{h_t} - 1 \right)^n H \left( \frac{h}{h_t} - 1 \right) - V_s \frac{\phi}{h}, \tag{7}$$

where we introduce the mass of suspended sediment per unit area, $\phi = hC$. Neglecting the advective term simplifies the model considerably : it reduces it to the ordinary differential equation (7). The latter describes how the concentration of suspended sediment evolves in response to the water level $h(t)$, which acts as a forcing function. To compute the sediment concentration, we use the ODE function from SciPy (Virtanen et al., 2020), to solve numerically the equation (7). This procedure yields the mass of suspended sediment per unit of bed area, $\phi$, from which we deduce the concentration $C = \phi/h$.

Our model reduces the dynamics of suspended sediment transport to an exchange of particle between the bed and the river, driven by the water level. As the latter must rise before the river can entrain any sediment, our theory can only produce counterclockwise loops of the concentration vs water-level relation. Finally, we did not establish equation (7) on a rigorous physical ground, but derived it from phenomenological considerations. Equation (7) should thus be considered as an ansatz, whose validity will depend on its ability to represent field data.

Equation (7) involves four parameters: the characteristic entrainment rate $\epsilon$, the threshold water-level $h_t$, the settling velocity $V_s$, and the exponent $n$. To determine the ability of our model to reproduce field measurements, we must find the values of these four parameters that best fit the data. In this aim, we use the optimization procedure described in the next section.

## 4  Application of the model to the Capesterre river

### 4.1   Model adjustment procedure

To test the model against field data, we select a time period, and extract the corresponding data. The latter form a collection of $N$ discrete values of concentration, $C_i^d$, and water level $h_i^d$, measured at times $t_i$, with $i = 1, .., N$. We then solve numerically equation (7), using the water-level data, $h_i^d$, as the forcing function. The resulting numerical solution depends on the values of the four parameters $\epsilon$, $h_t$, $V_s$, and $n$.

To compare it to the field data, we compute the theoretical concentration, $C_i^m$, for each of the $N$ discrete times $t_i$, and estimate the distance between the model and the data from the chi-square function,

$$\chi^2 = \frac{1}{N - n_p} \sum_i (C_i^d - C_i^m)^2 \tag{8}$$

where $n_p = 4$ is the number of parameters in our model.

To achieve the best fit between the model and the data, we need to determine the values of the parameters that minimize the chi-square function. For this purpose, we use the Trust Region Reflective algorithm method, implemented in the scipy.optimize.curve_fit function, and used in the model fitting wrapper of LMFIT (Non-Linear Least-Squares Minimization And Curve Fitting for Python) python library (Newville et al., 2016).

This optimization procedure is sensitive to the duration of the time period over which we apply it. In the next section, we discuss the calibration of our model on field data, over a short period of time corresponding to a single flood event. The case of longer time periods, which encompass several floods events, will be discussed in section 4.3.

### 4.2   Calibration of the model on a single flood event

Our model can only produce counterclockwise loops of the concentration vs water-level relation. We therefore test it first on the three floods of figure 2, which exhibit such loops. To do so, we set the initial values of the parameters to $\epsilon = 3$ mg m$^{-2}$ s$^{-1}$ for the entrainment rate, $V_s = 0.001$ m s$^{-1}$ for the settling velocity, $h_t = 15$ cm for the threshold water-level, and $n = 1$ for the exponent of the entrainment law. We then apply the optimization procedure described in the previous section, and determine the parameters that best fit the data (Table 1). Despite its simplicity, the model reproduces surprisingly well the evolution of the concentration of suspended sediment measured in the field (Fig. 2).

A plot of the theoretical concentration as a function of the water-level reveals that the model also accounts reasonably well for the hysteretic loop of the concentration vs water-level relation (Fig. 2, right). For the three flood events that we analyze, the model better represents the recession than the rise of the concentration. This may result from a bias of the optimization procedure which favors the recession limb, as the latter contains a greater number of data points than the flood rise.

The best fit parameters vary from one flood to the other (Table 1). The threshold water-level thus ranges from $h_t = 15$ to 23 cm, in agreement with the estimate based on a direct visualization of the data (Fig. 2). The settling velocity $V_s$ varies between $V_s = 6.2 \cdot 10^{-4}$ and $1.39 \cdot 10^{-3}$ m s$^{-1}$. As discussed later, these values are compatible with the settling velocity of quartz grains of size between 10 and 100 $\mu m$. The entrainment rate $\epsilon$ ranges between 0.641 and 21.1 mg m$^{-2}$ s$^{-1}$. We are not aware of any direct measurement of the rate of

entrainment of fine particles from a sediment bed, and cannot therefore assess whether these values are realistic or not. Finally, the exponent $n$ varies between 1.1 and 2.3.

Out of curiosity, we also tested our model against the flood of figure 3, for which the concentration vs water-level relation forms a clockwise hysteresis. As expected, the model utterly fails to represent the concentration of suspended sediment during this flood : it underestimates the concentration peak by about 45% and the concentration vs water-level relation is a mere line, instead of an hysteresis.

To summarize, equation (7) correctly models the transport of suspended sediment during floods, provided that the latter presents a counterclockwise loop of the relation between concentration and water-level. Yet, the best-fit parameter values vary from one flood to the other. This raises the question of the ability of the model to represent the evolution of the concentration over a long time, with a unique set of parameters. We address this problem in the next section.

### 4.3 Calibration of the model on a series of successive floods

To test the ability of equation (7) to represent sediment transport over a time longer than the duration of a single flood, we select a period of 4 days, marked by the occurrence of 4 successive floods, all of them characterized by a counterclockwise loop (Fig. 5). We then apply the optimization procedure, and determine the parameters that best fit the data (Table 1). This unique set of parameters reproduces reasonably well the evolution of the concentration during the four successive floods. The agreement is however not perfect : the model, for example, underestimates the amplitude of the second concentration peak by about 30%, and overestimates the amplitude of the last one by 34% (Fig. 5).

To better assess the quality of the fit, we compute the mass of suspended sediment exported from the catchment by integrating the sediment flux over the entire duration of our time series, $M = \int C\, q\, dt$, where $q$ is the flow discharge. By integrating the data, we obtained a mass of $M = 1.239 \cdot 10^3$ kg. The same calculation, conducted with the model, yields $M = 1.177 \cdot 10^3$ kg, that is about 5% smaller than the data. The model therefore provides a reasonable estimate of the mass exported out of the catchment.

Despite this encouraging result, we note that the model better predicts the data when its parameters are optimized on a single flood event, rather than on a series of floods. In short, the best-fit parameter values change from one flood to the next. This raises the question of their physical meaning, a topic we discuss in the next section.

## 5 Discussion

### 5.1 Physical meaning of the model parameters

The transport of suspended sediment depends on the properties of the sediment, on the specifics of the flow, and on the configuration of the catchment and the river. In our model, these properties are all lumped into the four parameters of equation (7). A change in the value of these parameters from one flood to the next therefore reflects a change of these properties.

The characteristic entrainment rate $\epsilon$, the exponent $n$, and the threshold of water level, $h_t$, for example, parameterize the expression of the rate at which particles are entrained from the bed (Eq. (4)). Any change of these parameters thus reflects a modification of the conditions in which fine sediment is entrained from the bed. Physics suggests that the entrainment rate depends on the threshold of grain entrainment, on the sediment availability, and on the structure of the sediment bed. In our model, the threshold of water level, $h_t$, accounts for the threshold of grain entrainment. We therefore hypothesize that the characteristic entrainment rate $\epsilon$ and the exponent $n$ might be related to the sediment availability and the organization of the sediment bed . Explicitly formalizing this relation is, however, a difficult problem. Instead, we now turn our attention to the settling velocity $V_s$ and the threshold water-level $h_t$.

### 5.1.1 Settling velocity

Our model represents the decline in sediment concentration after the flood peak with a single settling velocity, $V_s$. In the Capesterre river, the concentration of suspended sediment is always small enough to neglect interactions between particles, and the settling velocity primarily depends on the grain size (section 3). It is therefore tempting to try to infer the size of the suspended sediments from the value of $V_s$ deduced from the adjustment of the model to field data. To do this, we consider the suspended load as a mixture of homogeneous particles of identical size $d_s$ and density $\rho_s$, settling in water of density $\rho = 1000$ kg m$^{-3}$ and kinematic viscosity $\nu = 10^{-6}$ m$^2$ s$^{-1}$.

Following Abraham (1970), we write the drag force exerted on a particle as

$$F_d = \frac{\pi}{8} C_d \rho d_s^2 V_s^2, \tag{9}$$

where $C_d$ is a drag coefficient. The latter depends on the settling velocity through the Reynolds number $\mathrm{Re} = V_s d_s / \nu$ :

$$C_d = \left[ C_\infty^{1/2} + \left( \frac{24}{\mathrm{Re}} \right)^{1/2} \right]^2, \tag{10}$$

where $C_\infty$ is a constant that is about 1 in the case of natural grains (Andreotti et al., 2013). Equations (9) and (10) provide

| Fig. | Catchment | Period | $I_H$ | $\epsilon$ [mg m$^{-2}$s$^{-1}$] | $h_t$ [cm] | $n$ | $V_s$ [m s$^{-1}$] | $d_s$ [$\mu$m] |
|---|---|---|---|---|---|---|---|---|
| 2a | Capesterre | 05/09/19 19h - 06/09/19 10h | -0.194 | $21.1 \pm 0.1$ | $17.3 \pm 0.1$ | $1.580 \pm 0.008$ | $1.385 \pm 0.006\ 10^{-3}$ | 40.50 |
| 2b | Capesterre | 22/07/19 22h - 23/07/19 22h | -0.236 | $16.12 \pm 0.08$ | $22.5 \pm 0.1$ | $1.122 \pm 0.005$ | $7.54 \pm 0.08\ 10^{-4}$ | 29.39 |
| 2c | Capesterre | 10/03/21 10h - 11/03/21 24h | -0.278 | $0.641 \pm 0.001$ | $15.02 \pm 0.03$ | $2.300 \pm 0.003$ | $6.22 \pm 0.01\ 10^{-4}$ | 26.59 |
| 3 | Capesterre | 15/01/21 7h - 16/01/21 24h | +0.312 | $10^{-12}$ | $0.34 \pm 0.03$ | $5.23 \pm 0.08$ | $0.26 \pm 0.02$ | $\oslash$ |
| 5 | Capesterre | 06/02/21 18h - 10/02/21 23h | -0.168 | $23.2 \pm 0.09$ | $20.0 \pm 0.1$ | $2.031 \pm 0.007$ | $9.71 \pm 0.06\ 10^{-3}$ | 125.43 |
| 7a | Draix-Laval | 29/05/16 6h - 29/05/16 11h | -0.044 | $40.6 \pm 0.1$ | $7.22 \pm 0.03$ | $0.876 \pm 0.004$ | $3.12 \pm 0.01\ 10^{-3}$ | 63.29 |
| 7b | Draix-Laval | 29/05/16 15h - 29/05/16 20h | -0.249 | $15.3 \pm 0.1$ | $10.5 \pm 0.01$ | $0.500 \pm 0.004$ | $5.08 \pm 0.07\ 10^{-4}$ | 23.94 |

**Table 1.** Values of the best-fit parameters, $\epsilon, h_t, n, V_s$ and hysteresis index $I_H$ of the floods displayed on figures 2, 3, 5 and 7. $d_s$ is the grain size inferred from the settling velocity $V_s$, based on equation (12). $\oslash$ indicates irrelevant values ($I_H > 0$).

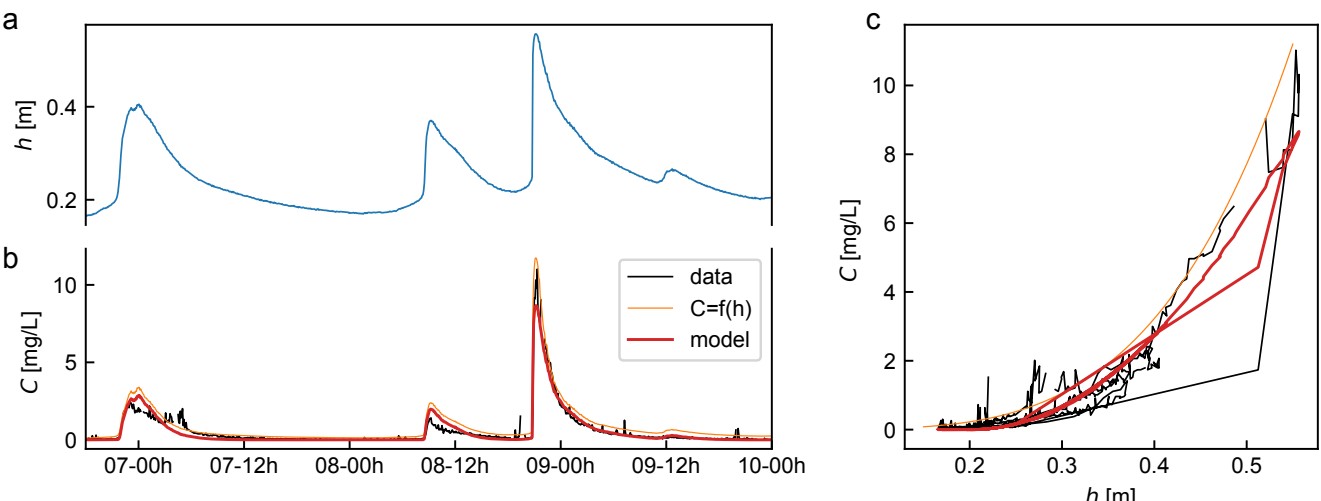

**Figure 5.** (a) Water level (blue line) and (b) concentration of suspended sediment in the Capesterre river between 06/02/2021, 18:00 and 10/02/2021, 23:00. (c) Concentration of suspended sediment vs water level. Black line : concentration measured from the turbidity. Orange line: concentration calculated from the flow depth, based on an empirical rating curve. Red line: concentration predicted from the best-fit model.

a general expression of the drag coefficient that represents reasonably well the drag force from $Re = 0$ to $Re \approx 5000$ (Abraham, 1970). In particular, at small Reynolds number, the drag coefficient simplifies into $C_d = 24/Re$, and one recovers the classical Stokes formula, $F_d = 3\pi\mu d_s V_s$. Conversely, for high Reynolds numbers, the drag force reduces to $F_d = \frac{\pi}{8} C_\infty \rho d_s^2 V_s^2$, as expected in a turbulent flow.

Balancing the drag force with the reduced weight of the particle, $F_g = (\pi/6)(\rho_s - \rho)gd_s^3$, we obtain the following expression for the settling velocity,

$$V_s = 6\frac{\nu}{d_s}\left[\left(\sqrt{\frac{(\rho_s - \rho)gd_s^3}{27\rho\nu^2} + 1}\right)^{1/2} - 1\right]^2, \quad (11)$$

where $g$ is the acceleration of gravity. Inverting this equation yields the grain size as a function of the settling velocity:

$$d_s = \frac{24\,\nu}{\left[\left(\sqrt{32\frac{(\rho_s - \rho)g\nu}{\rho V_s} + \frac{V_s}{4}}\right)^{1/2} - \left(\frac{V_s}{4}\right)^{1/2}\right]^2}. \quad (12)$$

The best we can hope for is that equation (12) provides a value of the grain size that is representative of the suspended load. Indeed, our model assimilates the suspended load to a mixture of homogeneous particles of identical size. In reality, suspended load often consists of particles of different sizes and densities, which settle at different velocities. The conversion of settling velocity into a grain size only works as long as the size distribution of the grains in suspension is narrow.

Moreover, our model assumes that turbulence is high enough to homogenise the concentration of suspended sediment in the water column. In fact, laboratory experiments show the existence of a vertical gradient of suspended-sediment concentration, even when the bed roughness is of the order of the flow depth (Grams and Wilcock, 2007). If such a concentration gradient forms in our river, the model prediction will likely overestimate the settling velocity by a factor equal to the ratio of the near-bed concentration to the averaged one. In this case, equation (12) underestimates the grain size.

Keeping these limitations in mind, we use equation (12) to estimate the grain size in the Capesterre river. The adjustment of our model to the field data yields the settling velocities reported in Table 1. Setting the density of sediment to $\rho_s = 2700$ kg m$^{-3}$, we use equation (12) to turn these values into grain sizes. We find that the latter falls between 26 and 125 µm (Tab.1), in agreement with the range of suspended-sediment sizes reported in the literature (Sheldon et al., 1972; Wilcock et al., 2009).

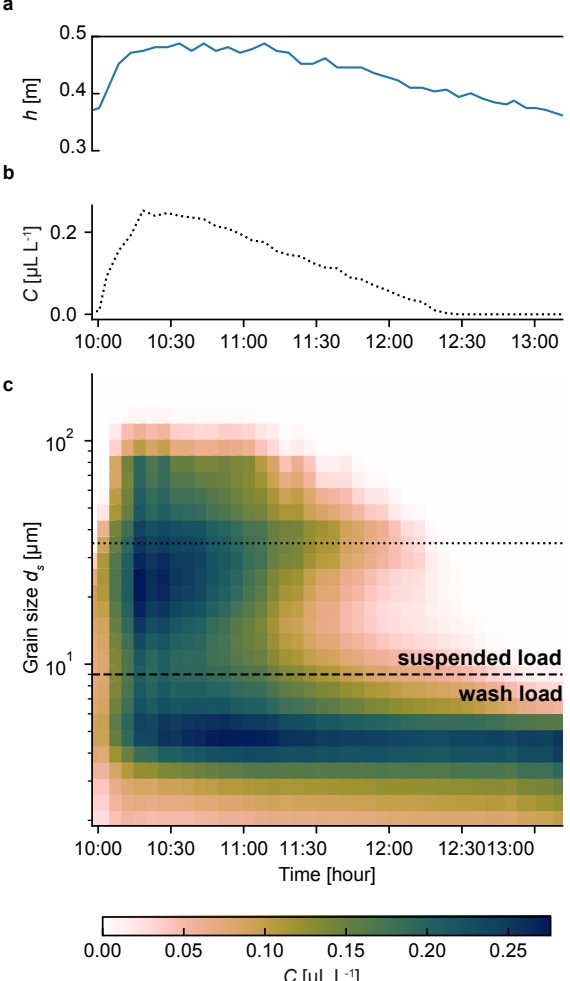

**Figure 6.** Data acquired with a LISST-Streamside in the Capesterre river, on October 30th 2010. (a) water level and (b) concentration of particles of median size 34.8 µm (class sizes between 31.9 and 37.7 µm) as a function of time. (c) Evolution of the concentration (color code) as a function of time (horizontal axis) for each of the 32 classes of grain sizes (vertical axis). Dashed line : transition between wash and suspended load. Dotted line : concentration profile displayed in (b).

For lack of simultaneous measurement of turbidity, grain size, and water-level in the Capesterre river, we cannot compare the grain size calculated from the model with that mea-

sured in the field. In 2010, however, the observatory Ob-sERA installed a LISST-StreamSide in the river, and kept it running for a few months. The LISST-StreamSide (Laser In-Situ Scattering and Transmissometry, Sequoia Scientific inc.) is a laser particle sizer which measures the concentration of suspended particles in 32 logarithmically spaced size classes from 2 to 381 µm (Agrawal and Pottsmith, 2000). Assuming that the sediment size has not drastically changed since 2010, we compare the grain sizes measured by the LISST to that obtained from a fit of our model to the three 2019 floods of figure 2. Figure 6 shows the evolution of the grain size distribution during a flood, on October 30th, 2010. We observe a smooth transition between two different behaviors, depending on the grain size. Below a diameter of about 10 µm, the concentration weakly depends on the water level: it is small, but not zero, before the flood, increases during the flood rise, relaxes after the flood peak, but does not return to zero over the duration of the recording. In this size range, particles settle very slowly on the bed, if at all. Based on these observations, we assimilate the particles of size less than 10 µm to the washload, although recent investigations suggest that this conceptualization of washload is over-simplified (Ren and Packman, 2007; Dallmann et al., 2020).

Above about 10 µm, the concentration of particles follows the evolution of the water level: it increases during the flood rise, and relaxes to zero after the flood peak (Fig.6b, c). We therefore interpret the corresponding population of particles as the suspended load. During the flood, their size ranges from 10 µm to 168 µm, a range that is consistent with the values calculated from the model. Encouraged by this result, we now turn our attention to the threshold water-level.

### 5.1.2 Threshold water-level

Our results shows that the flow entrains fine sediment from the bed when the water level exceeds a threshold of approximately $h_t \approx 20$ cm. Although this value is computed for only four floods, it is, by far, the most robust result of our manuscript: direct observation of the data confirm that the concentration of suspended sediment rises only if the water level exceeds this threshold of about 20 cm, independently of the magnitude of the flood or the time of the year (Fig. 1 e, f).

In our model, the water-level is a proxy for the shear stress $\tau$ that the river exerts on its bed (section 3). For sediment to be entrained by the flow, this stress must exceed a threshold $\tau_t$, a condition usually expressed in terms of the dimensionless Shields stress,

$$\frac{\tau}{(\rho_s - \rho)gd_s} > \theta_t, \tag{13}$$

where $\theta_t$ is the threshold Shields stress, a dimensionless number that varies with the grain size and the flow regime (Shields, 1936; Andreotti et al., 2013). The fourth parameter

of our model, the threshold water-level, should therefore be linked to the threshold Shields stress.

To determine this link, we first note that the width of the Capesterre river is constant over a few hundred meters upstream of the gauging station. There, assuming that the flow is uniform, we approximate the threshold shear-stress by

$$\tau_t = \rho g h_t S, \tag{14}$$

where $S$ is the slope of the river in the downstream direction. The latter, estimated from a digital elevation model (spatial resolution : 5 meters), is about $S = 6.3 \cdot 10^{-2}$ at the La Digue station. Combining equations (13) and (14) yields the relation between the threshold water-level and the threshold Shield-stress,

$$h_t = \frac{(\rho_s - \rho)\, d_s\, \theta_t}{\rho\, S}. \tag{15}$$

In section 5.1.1, we found that the size of the suspended particles varies between 26 and 125 μm. The corresponding threshold Shields stresse ranges from $\theta_t \approx 0.1$ for the largest grains, to $\theta_t \approx 0.2$ for the smallest ones, which are sensitive to cohesion forces (Shields, 1936; Van Rijn, 1984; Andreotti et al., 2013; Dunne et al., 2022). The threshold water-level, deduced from equation (15), thus varies between $h_t \approx 10^{-3}$ and $h_t \approx 10^{-2}$ cm. These values are inconsistent with the threshold $h_t \approx 20$ cm estimated from the model.

As the bed of the Capesterre river is covered with centimeter-size sediments (Fig. 1 b), we suspect that this inconsistency might be the signature of bed armoring, an effect commonly observed in gravel bed rivers as well as in aeolian systems (Ferdowsi et al., 2017b; Gao et al., 2016). The river bed is said to be armored when a layer of coarse sediment overlies finer material, preventing it from being entrained in the flow, unless the armoring particles move first (Misset et al., 2021). If this scenario holds, the threshold of suspended sediment transport should coincide with that of the coarse particles. Measurements of the grain-size distribution of the bed of the Capesterre river, at the La Digue station, indicate that the river bed is predominantly made of gravels and pebbles, with a median grain size $d_{50} \approx 10$ cm. The corresponding threshold Shields stress is $\theta_t \approx 3 \cdot 10^{-3}$, for which equation (15) predicts a threshold water-level $h_t \approx 10$ cm, comparable to the value deduced from our model. We therefore conclude that, in the Capesterre river, the threshold of suspended sediment transport is set by that of the coarse particles.

This result does not imply that fine sediments are stored in the subsurface, under a layer of coarse sediment. In fact, in the Capesterre river, silt and sand particles form tiny patches at the surface of the bed. These patches are trapped in the narrow space between neighboring pebbles. As the bed roughness is high (coarse sediment have a median size is about 10 cm), silts and sands are effectively screened from the flow, and remain trapped between coarse particles until the latter start moving.

## 5.2   Application of the model to a small alpine catchment

Despite its simplicity, our model accounts for the transport of suspended sediment during floods, provided that the relation between concentration and water-level forms a counterclockwise loop. We also found that two parameters of the model, the settling velocity and the threshold water-level, can be used to estimate the size of the suspended sediment and to detect possible armoring of the bed. So far, however, we have only tested the model on data acquired in the Capesterre River. To test its versatility, we now apply it to the description of suspended-sediment transport in a completely different geological context, that of the Laval catchment.

The Laval catchment, located in the southern part of the French Alps is a small catchment of area 0.86 km², drained by the Laval stream (Fig. 7). Its elevation ranges from 850 to 1250 m a.s.l, and the annual rainfall rate is about 900 mm yr$^{-1}$, with heavy rainfall events during spring and summer, and less intense but longer rainfalls in autumn. Besides, this catchment can also experience snowfalls during winter (Ariagno et al., 2022). Unlike Capesterre, vegetation covers only 32% of the Laval catchment (Carriere et al., 2020) (Fig.7a). The latter is underlain by easily erodible Middle Jurassic black marls, leading to the formation of steep badlands slopes within the catchment (Ariagno et al., 2022). A gauging station operated by the Draix–Bléone Critical Zone Observatory monitors the concentration of suspended sediment and the water level in the stream. The corresponding data are available on the Draix–Bléone Critical Zone Observatory website.

Figure 7 displays the water level and the concentration of suspended sediment during two floods, extracted from the catalog of the Draix–Bléone Critical Zone Observatory. The water level ranges from 5 to 25 cm, and the concentration reaches up to 35 g L$^{-1}$, that is about 1000 times higher than the maximum concentration measured in Capesterre. During these two floods, the concentration vs water-level relation forms a counterclockwise hysteretic loop, which makes them, in principle, compatible with our model (Fig.7b and c, right panels).

Encouraged by this observation, we apply the optimization procedure introduced in section 4.1, and determine the parameters that best fit the data (Tab. 1). As in Capesterre, the model represents rather well the evolution of the concentration of suspended sediment measured in the field (Fig.7 b, c) with similar $\chi^2$ values (< 0.04). Setting aside the exponent $n$, we note that the best-fit parameter values fall within the same ranges as those obtained in Capesterre (Tab.1).

A comprehensive evaluation of the model would require a systematic test against data measured not only in Draix-Laval, but in several others catchments. Such a work is beyond the scope of the present paper. For the time being, we simply note that the model seems able to reproduce sediment transport in a context different than that of Capesterre.

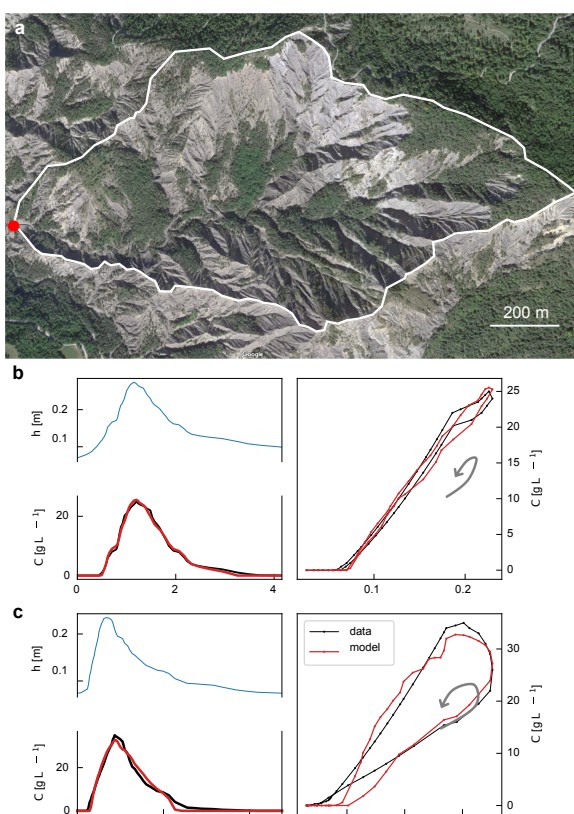

**Figure 7.** Floods in the Draix-Laval catchment. (a) Aerial view and boundaries of the catchment (© Google map), red dot locates the outlet where the Laval station is (44°08'26.7"N 6°21'39.4"E). (b) and (c) two floods recorded on 06/29/2016, from 6:00 to 11:00, and from 15:00 to 20:00, respectively. Left panels: time series of the water level (blue line) and the concentration of suspended sediment (black line) measured at the gauging station. Right panels : relation between concentration and water-level. Gray arrows indicate the direction of the hysteresis loops. On each panel, the red line is the concentration predicted from the best-fit model.

## 5.3 Strengths and limitations of the model

In this manuscript, we adopt a systemic approach : instead of focusing on details, we treat the river as a uniform reservoir of homogenous sediment, and reduce the dynamics of suspended sediment to an exchange of particles between the bed and the flow, driven by the water level. The resulting model describes the transport of suspended sediments by means of an ordinary differential equation, while the characteristics of the sediment and the river – such as the grain size distribution, the availability of sediment, or the threshold shear stress necessary to set sediment in motion – are all lumped into four parameters : a representative settling velocity, a threshold water-level, a characteristic entrainment rate, and a dimensionless exponent.

Despite its simplicity, the model represents reasonably well the transport of suspended sediment in two small catchments, Capesterre and Draix-Laval, provided that the relation between concentration and water level forms a counterclockwise loop. The simple exchange of particles between the bed and the flow may therefore suffice to form these loops, which are not necessarily the signature of a particular configuration of sediment sources, or of a difference in speed between the flow and the flood wave.

In many catchments, the sediment yield of the river is calculated from the flow stage – or, equivalently, the flow discharge – based on a sediment rating curve (Bierman and Montgomery, 2014). In contrast to our model, this approach assumes that the sediment concentration is a one-to-one function of the water stage, and cannot represent hysteretic loops of the relation between concentration and water level. Yet, out of curiosity, we compare the prediction of our model with the concentration estimated from the sediment rating curve deduced from direct measurements of the concentration of suspended sediment in the Capesterre river. For the series of floods displayed in Figure 5, the rating curve systematically overestimates the concentration measured by the turbidity sensor. As a result, the sediment yield calculated from the rating curve exceeds the one measured with the turbidimeter by about 54%. By comparison, the prediction of the model fall within 5% of the data. These observations do not necessarily disqualify the use of the sediment rating curve. Indeed, the latter is constructed from samples collected by an automatic water sampler. The latter collects 24 samples, one every 15 to 60 minutes, as soon as the river stage exceeds a threshold set by the operator. In the dataset at our disposal, this threshold varies between 30 and 80 cm, that is way above the threshold of entrainment in the Capesterre river. As a result, most concentration data belong to the falling rather than to the rising part of the hysteresis. This biais likely explains why, in our case, the rating curve overestimates the concentration of suspended sediment in the river. Unfortunately, we do not have enough measurements of the concentration to correct this bias.

By construction, our model can only underestimate the sediment load during floods that develop a clockwise relation of concentration vs water level. Yet, even during such an event, the sediment yield calculated from the rating curve exceeds the one measured with the turbidimeter by about 32%, while the prediction of the model fall within 3% of the data (Fig. 3).

Obviously, our model is far too simple to incorporate all the processes involved in the transport of suspended sediment. Yet, simplicity has its advantages. First, our model relies on only four parameters. As a result, its computational cost is very small, compared to the one required by more complex approaches, such as hydromorphodynamic or distributed models (Vercruysse and Grabowski, 2019; Lepesqueur et al., 2019). Indeed, the calibration of our model against a single flood, such as the ones of figure 2, only takes

a few minutes of computational time. This small number of parameters also reduces the risk of data overfitting, which, sometimes, hinders the use of more complex models (Gao, 2008).

Secondly, each parameter of our our model admits a physical interpretation : the representative settling velocity is related to the size of the suspended sediment; the threshold water-level acts as a proxy for the threshold shear stress; the characteristic entrainment rate and the dimensionless exponent parameterize the ability of the flow to entrain sediment out of the bed. One may therefore use these parameters to infer information about the properties of the catchment, however imperfect this information might be. In the Capesterre river, for example, the values of the parameters calculated from a fit of the data to our model change from one flood to the next, probably reflecting changes of the river and sediment properties. Changes of the settling velocity $V_s$, for example, reflect changes of the size distribution of the suspended sediment.

Last, but not least, the calibration of our model requires a single field station. This makes it particularly suitable for the analysis of data acquired in small catchments, as the latter are often monitored with a single station at the basin outlet (Gaillardet et al., 2018).

Simplicity comes at a cost, and our model suffers from many limitations. First, it is unable to represent clockwise loops of the concentration vs water-level relationship. In the Capesterre river, such events amount to about 40% of the floods, and are therefore far from anecdotal. In practice, this inability to account for clockwise loops limits the use of the model to short time intervals, over which the concentration/water level relationship remains counterclockwise.

Based on field observations, we suspect that, in the Capesterre river, clockwise loops form when the stock of fine particles stored in the sediment bed is too small. In this supply limited configuration, the entire stock of fine sediment is rapidly entrained by the flow, and the concentration reaches its maximum prior to the hydrograph peak (Williams, 1989). We could incorporate this effect into our model by introducing a second equation, that accounts for the quantity of fine sediment stored in the bed. In this improved model, the rate at which the flow entrains sediment from the bed would depend not only on the water level, but also on the quantity of fine sediment available on the bed. Such an improvement is a work in progress.

Equation (7) assumes that the contribution of the advective term is negligible. This assumption, which considerably simplifies the model, holds as long as the representative settling time of the particles is short compared to the duration of the flood, $h_t/V_s \ll \tau_f$ (section 3). This condition depends on the size of the suspended sediment and on the river depth. It is satisfied in the Capesterre river. It might not be the case in other rivers.

As all theories, our model is susceptible to predict the correct result for the wrong reasons. In lowland rivers, for ex-

ample, the slope is shallow and flood waves may travel faster than the particles suspended in the flow. The resulting concentration vs water-stage relation then forms a counterclockwise loop. We suspect that a fit of our model to such an event would correctly represent the data. In this case, however, our model, which does not incorporate the propagation of flood waves, would be right for the wrong reasons, and the parameter values would be meaningless. Such a configuration requires more sophisticated equations that account explicitly for the flow (Lepesqueur et al., 2019). Our model should thus be used with caution, and its parameters be interpreted with care.

Finally, the model that we propose is far too simple to incorporate all the mechanisms that may influence the transport of suspended-sediment. In particular, it assimilates the suspended load to a suspension of grains of uniform and constant size and density. Yet, recent investigations suggest that flocculation and complex interaction with the bed may change the size distribution of fine sediment over the course of a flood (Ren and Packman, 2007; Dallmann et al., 2020; Lamb et al., 2020). As a matter of fact, the chemical composition and the size of the particles suspended in the Capesterre river change with time : at the beginning of a flood, the suspended load is dominated by litter debris and Allophanes; sand particles appear later, when the flow discharge is sufficiently high (C. Dessert, *private communication*). Modeling such a complex behavior is a difficult problem, far beyond the abilities of the model presented in this work.

## 6   Conclusions

Based on data acquired in the Capesterre river, a small tropical river in Guadeloupe, we develop a phenomenological model that describes the transport of suspended-sediment. Instead of focusing on details, our model adopts a systemic approach, and treats the river as a uniform reservoir of homogenous sediment entrained by the flow. It accounts neither for the propagation of a flood wave nor for the consequences of an inhomogeneous sediment source. In addition, the model assumes that the entrainment of sediment is not limited by supply, a condition often referred to as "transport limited". In short, we reduce the transport of suspended sediment to a uniform exchange of particles between the bed and the flow, driven by the water level.

Despite these simplifications, the model represents reasonably well the transport of suspended sediment in two small catchments, Capesterre and Draix-Laval, provided that the relation between concentration and water level forms a counterclockwise loop. The model, however, cannot represent clockwise loops of the concentration vs water-level relationship. This restricts its use to short time intervals, during which the concentration vs water level relationship remains counterclockwise.

Our model describes the transport of suspended sediments by means of an ordinary differential equation, while the characteristics of the sediment and the river – such as the grain size distribution, the availability of sediment, or the threshold shear stress necessary to set sediment in motion – are all lumped into four parameters: a settling velocity related to the size of the suspended sediment, a threshold water-level which acts as a proxy for the threshold shear stress, a characteristic entrainment rate and a dimensionless exponent, which are related to the sediment availability. This simplicity gives the model a low computational cost. In addition, calibration of the model requires a single field station. This makes it particularly suitable for the analysis of data acquired in small catchments, as the latter are often monitored with a single station at the basin outlet (Gaillardet et al., 2018).

In the Capesterre river, the value of the parameters deduced from a fit of the model to the field data, changes from one flood to the next, probably reflecting changes of the river and sediment properties. Changes of the settling velocity $V_s$, for example, reflect changes of the size distribution of the suspended sediment. Similarly, the threshold water-level, $h_t$, is a proxy for the threshold stress necessary to entrain sediment from the bed. As for the entrainment rate $\epsilon$ and the exponent $n$, we suspect that they are linked to the availability of fine sediment in the river bed. If these conjectures prove to be correct, our model might offer the opportunity to detect river-wide changes, like, for example, modifications of the sediment size and availability induced by bank incision or landsliding.

**Author contributions.** A.R-B. drove the science, wrote the model and produce the figures. A.L. and E.L. initiated the research. E.L. provided the original concept of the model. A.L. provide his expertise in inverse problem and modeling. E.G. provided the background on erosion in tropical environment. P.A. and C.D. provided data and expertise that support this research. All authors contributed to the paper writing and sharing ideas.

**Competing interests.** Authors declare no competing interests

**Acknowledgements.** The data used in this paper were collected by two observatories of the OZCAR research infrastructure : the Observatoire de l'Eau et de l'éRosion aux Antilles (ObsERA, INSU-CNRS, http://webobsera.ipgp.fr/), located in Guadeloupe, and the Draix–Bléone Critical Zone Observatory (https://bdoh. irstea.fr/DRAIX/), located in the french Alps. These data are freely available from the website of each observatory. Authors thank the anonymous referees and associate editor for their insightful review that help improving the manuscript. Author thank the members of ObsERA, and specially T. Kitou, V. Robert, O. Crispi, J. Ajax and B. Lamaille, for their technical support. We are also grateful to C. Le Bouteiller for sharing her knowledge and expertise on the Laval catchment. A.R.B. and A.L. acknowledge the support from LabEx UnivEarthS (ANR-10-LABX-0023 and ANR-18-IDEX-0001).

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
