# Peer review of "Phenomenological model of suspended sediment transport in a small catchment"

_Earth Surface Dynamics, 2022_

## Referee Comment (RC2)

Title : Phenomenological model of suspended sediment transport in a small tropical catchment

**General comments**

The authors present a new model to simulate suspended sediment concentration based on water depth time series, for floods that exhibit a counterclockwise hysteresis pattern between concentration and discharge. After calibration on the Capesterre tropical catchment, the model appears to be able to simulate correctly such hysteresis patterns, and the concentration values. Moreover, the calibrated parameter values are consistent with physical explanations of river processes. The model is also tested against another catchment where it provides consistent results.

The paper addresses the issue of suspended sediment modeling, which is clearly relevant to the scope of SURF. The proposed modeling approach is novel, yield convincing results, and will certainly of great interest for the community. The paper is clearly written. My main concern is that I find this paper is not well rooted into the literature, particularly concerning suspended sediment modeling. This should be improved both for the introduction and discussion sections (see details below) in order to make the paper stronger. Moreover there are several points that deserve more accurate explanations and discussion (the concentration-turbidity relation, the model assumptions, the quantification of model performance, see line by line comments) and some conclusions that are a bit too quick. I therefore recommend major revisions before accepting this article in ESURF journal.

Introduction: I find that the authors do not relate their work to existing literature on suspended sediment concentration or yield models. There is a large body of research concerning the modeling of suspended sediment yield, with either advection-settling-erosion equations, concentration-discharge relations, empirical models at various time-scales …. Therefore, it would be useful for the reader to provide some background on existing approaches, explain their limitations and show how the proposed approach brings novelty and improvements. I am totally convinced that it does bring novelty but this has to be explained in relation to the literature.

Discussion: Currently, the main discussion is about the meaning of the parameters (section 5). Even if this is a very interesting section, I think that the paper deserves a larger discussion section, including (3) a discussion of what this new model brings in comparison to existing models of suspended sediment yield, (4) a discussion on the limitations of this model, in particular concerning the validity of the assumptions, and the uncertainty related to the concentration-turbidity relation, and (5) some perspectives and applications of such model, for instance at larger time and spatial scales. I suggest adding a full discussion section to the paper, which could incorporate the existing sections 5 (meaning of parameters) and 6 (test against another data set), and address all the other points mentioned above.

Structure: I might be a bit conservative but I am not totally convinced by the current structure of the paper. It reads well but I find that it does not make clearly visible what the results of the paper are. Is the field data from Capesterre a result from this paper, or a preliminary summary of previous studies, only presented here as an introduction for the modeling work? I would suggest using a more standard organization with a methodological section (field setting, field data acquisition, model construction and calibration), a result section (field data sets and calibration results) and a full discussion section that could address all the points mentioned earlier. This is not a critical point for publication however and I would let the editor comment on the necessity, or not, to change this structure.

**Line by line comments**

Line 2: "the model correctly represent the transport of suspended sediment": I think this sentence is not exact; the model actually does not represent the transport of sediment itself. I suggest replacing by "the model correctly represents the evolution of sediment concentration at the outlet"

Line 6: "both of which are related to the availability of fine sediment": I don't think that this is demonstrated in the paper. I suggest removing this from the abstract.

Line 47: Perhaps worth mentioning that the distance of sediment sources to the outlet is an important factor of hysteresis. And that this effect can be coupled with the spatial heterogeneity of rainfall.

Line 51: Here I think that the authors should review existing approaches for modeling suspended sediment transport. There had been a lot of research on this topic so they should at least explain what the limitations of existing approaches are and why they choose to build a new type of model.

Line 94 and figure 1d: The calibration of the turbidity-concentration relationship is quite noisy. Could the author introduce an estimation of the uncertainty for the concentration extracted from this relation?

Moreover, is there any information available on the variability of suspended sediment grain-size during the floods? It is sometime argued that hysteretic patterns in concentration may result from changes in the turbidity-concentration relationship due to grain-size variability (Landers and Sturm, 2013, https://doi.org/10.1002/wrcr.20394). Figure 1d suggests that there is a great variability in this concentration-discharge relationship, which could be related to a variability in grain-size. **This is an important point to check before attributing the hysteresis in turbidity to a hysteresis in concentration, which is the focus on the paper.**

Update: having read the whole paper and seen figure 6, it seems that there is indeed a shift in grain-size during the flood. I suggest that the authors use this data to estimate a mean grain-size on the rising and falling limb, then search for information from the sensor manufacturer, and/or from the literature, to estimate if this grain-size shift could affect, and how much, the turbidity-concentration relation of Figure 1d.

Line 95: Is the field data on water level and concentration already published somewhere, or could it be published with this paper? I find it a little frustrating to read about a 3 year data set with 217 floods without seeing more than what is displayed in Figure 1e, 2 and 4. Even if I trust the authors concerning their analysis on flood extraction and hysteresis calculations, I think it would be a good thing to make this data set available, both for the reader, and for the data producers themselves.

Figure 2: The three floods that are selected here have a very simple shape, with an abrupt increase and an "exponential-like" decline. Are these floods representative of the whole data set, do most floods exhibit the same shape in this catchment? Are there any multipeak events?

Line 145-147: Do you think that this bed replenishment from hillslope really occur gradually, with no effect during floods? I would expect that hillslope processes such as landslides or debrisflows should be more frequent during strong rainfall, therefore could also represent sediment sources during floods on top of the riverbed source itself.

Line 148: If available, an information on suspended sediment grain-size would be useful here. If it is mostly silt, rather than sand, it would reinforce this assumption

Line 152-155: **This is indeed a very strong assumption, which is not realistic at all to me. It raises the issue of water mass conservation**. With a uniform rain, the amount of water arriving at one point should scale with drainage area. Therefore, it is not possible for the water depth to be always the same at every point in the system. Even if you argue that the river width increases downstream, this increases is generally thought to scale with drainage area to a power < 0.5 (see Lague 2014, DiBiase and Whipple, 2011 for instance) therefore this increase is not sufficient to conserve water unless water depth increases with drainage area too.

To my mind, this can not be presented as an "simplifying assumption driven by field observation" (line 140) since it does not seem to be rooted in any observations, physical principles, or expected behavior of the system.

Perhaps an alternative approach to simplify the model would be to write the full equation, then make the assumption that the advection terms are negligible (with no a priori physical basis), then check that this is indeed the case after solving the model. This could be done by comparing estimates of advective fluxes, and erosion and deposition rates after calibration.

Line 156: Removing the advection terms: does this imply that there is no flow velocity at all (which sounds problematic since bottom shear stress (induced from flow velocity) is responsible for sediment entrainment? Or is it that flow velocity and concentration are the same upstream and downstream therefore the advection contribution is null ? I understand that it should be the second option, but this should be made clearer to the reader.

Line 167: Perhaps you could detail that an increase of h implies an increase of the flow velocity hence of bottom shear stress?

Line 168: This form of erosion rate is not new, I think it is a classical form for erosion rate as a function of a bottom shear stress in the sediment transport literature. The author could acknowledge here that they build the model based on these existing approaches.

Line 204 and further: Why using "assimilation" instead of "calibration"? As far as I understand, the procedure described here is a model calibration?

Line 209-211: "the model reproduces surprisingly well" and "the model accounts reasonably well": **could you quantify how well does the model perform, using a quantitative index**? For instance, a RMSE, or other. This would be useful for comparison with the results over a longer period, and the results on the Laval catchment.

Line 215: Any suggestion to address this bias? Perhaps this could be discussed in a future discussion section?

Line 225: It would be interesting to compare the model results with the results from a simple rating curve approach (fitting a relation between Q and C), both at the scale of the event, and of the chronicle.

Line 230: is it possible to run the same test over a full year of data ?

Line 233: "reasonably well": can you quantify this?

Line 240: Would a model based on a simple rating curve perform as well, or better, over this period? The present model definitely brings some improvement in terms of simulating the hysteresis, but does it also improve the simulation of total sediment export compared to a simpler model? This could be quantified by calibrating a simple power law relation between discharge and concentration, calibrating this model against the same data and computing the same performance index (RMSE or other) than for the present model.

Line 248: "we suspect that the characteristic erosion rate and the exponent n reflect changes in the sediment availability": **Could you explain a bit more? Why? Any suggestion to test this hypothesis ?**

Line 261-270 : Here you use an expression for the drag force (equation 7) which is correct for a turbulent flow, but the particle Reynolds number ($Re_p = V_s d_s/\nu$) is smaller than 1. Therefore you should rather use the classical Stokes formula, $V_s = (s-1)gD^2/(18\nu)$.

(And I don't think that Andreotti et al (2013) were the firsts to introduce Stokes' law by the way ☺)

Line 290: Do you mean that the water level is a proxy of the bed shear stress?

Line 297: This equation (12) requires an assumption of flow uniformity, which is worth mentioning to the reader.

Line 301: What is the resolution of the DEM that is used to compute this slope? It is usually quite difficult to obtain accurate river slope values from a DEM unless the resolution is really fine. Moreoever, using equation 12 to estimate the shear stress requires that the flow is uniform at the gauging station, which is often not true (when stations are located on a weir or in a narrowing section for instance). Is it (approximately) the case at La Digue station?

Line 310-317: This is an interesting point to discuss, particularly in relation to Misset el al, GRL, 2021, that showed how bed mobility was able to release fine particles and increase concentration. **However, this result is in contradiction with the first assumption of the model (Line 144) according to which there was a large amount of fine sediment available in the riverbed.**

I understand that for a large flood, coarse bed sediment can be entrained therefore release fine sediment from the subsurface. But there should be a range of intermediate floods for which the coarse particles do not move. For such floods, either there is a large amount of fine sediment on the bed hence the critical shear stress should correspond to a small sediment size, either fine sediments are mostly stuck in the subsurface hence the transport-limited assumption does not hold. This should be discussed into more detail. Perhaps you could calibrate the model on floods of various amplitude and check if there is a different trend in the calibration for intermediate and high floods? Otherwise, I think it is a bit too early to conclude in a general way from these 4 values of ht that "the threshold of suspended sediment transport is set by that of the coarse particles"

Line 399: "the model represents rather well": What do you mean by "rather well"? Could you quantify? (see earlier comment)

Line 341: "the best-fit parameter values fall within the same ranges as those obtained in Capesterre". **So what does it mean? Could you discuss on the potential implications of this observation?** Does a similar range of falling velocity suggests that grain-sizes are the same? Does a similar range of threshold water level indicate that suspended sediment transport is also controlled by armoring in the Laval catchment?

Line 347: "that accounts for the transport of suspended sediment": I don't think that the model accounts for the transport (no advective terms). I suggest replacing by "that represents the temporal evolution of the concentration during floods.

Line 349: same comment as above

Line 356: This has only been suggested in line 248 but has not been discussed or demonstrated. I don't think that this should be part of the conclusions. "We suspect that" without any other kind of explanation is not a scientific result in my opinion.

Line 365-368: **I think these points deserve to be addressed in a discussion section**. In particular, a discussion concerning the limitations of the model, and the effects of grain-size, should be developed in the paper earlier than in the conclusions.

---

## Author Comment (AC1)

**1    Reviewer 1**

*This study provides an elegant semi-empirical model based on sediment entrainment and settling to explain certain suspended-sediment hysteresis loops in rivers. One of the ways this study is most useful is in providing a null model for understanding sediment hysteresis. It also generates testable predictions about the dominant grain size during floods and the role of sediment armoring. This is a clever study which I believe will be a significant contribution to the study of sediment hysteresis loops in rivers and fluvial sediment transport more generally.*

*My only major comment on the paper is that I felt that it needed a more thorough discussion. The way the paper is currently structured, small bits of discussion are sprinkled throughout and in particular the "conclusion" section, but a more thorough discussion of how this work fits within the broader context of the scientific field is warranted. Some questions I believe would be pertinent in a broader discussion include:*

- *(How) does this work alter the interpretations of previous studies? A better sense of how this work fits within the scientific literature would be useful.*

- *What are other mechanisms for generating counterclockwise hysteresis loops? Would this be reflected in a poor model fit, or might the model fit the loop well but for the wrong reasons? How might one check for the latter?*

- *What are some ways that this model could be tested? E.g. grain size measurements, bed characterization, etc.*

- *Are there certain types of river systems where this model can be expected to perform poorly?*

- *How might this model be used in the future to improve studies of hysteresis and better understand sediment transport dynamics? In other words, an explanation of why/how this is a significant scientific contribution.*

- *Could this model be used to improve predictions of sediment yield, e.g. by going beyond a simple sediment rating curve?*

- *Why do best-fit parameters (such as grain size) change flood to flood? Is the amount of change between sequential floods predicted by the model realistic? Given L152-156, do these changes reflect basin-wide changes? What are some mechanisms that could be driving this change?*

The reviewer's substantiated comments helped us improve the manuscript in many ways. Her/his main comment about the need of a more thorough discussion led us to substantial modifications of the manuscript. In particular, we have added to the introduction a series of paragraphs that review the different approaches for modeling suspended sediment transport. The new manuscript also contains a discussion section, where we try to address the questions raised by the reviewer.

*Would it be possible to show either on existing plots or as a new plot how this model performs compared to regressing the concentration as a function of discharge, i.e. a simple sediment rating curve? Would be useful to compare the two in the time series plots to highlight the improvement provided by this work.*

As requested, we now compare the performance of the model with the more classical rating-curve approach, over the series of floods of Fig. 5. We discuss the results of this comparison in sections 4.3 and 5.3.

*All figures: I find the upside down hydrographs a bit hard to read. Not sure if this was to help show the offset between peaks (which is still hard to see), but perhaps a better way to do that is to add a vertical dashed line corresponding to the peak discharge.*

We modified all figures according to the referee's comment.

*L119 – Should the denominator be $h_{max} - h_{min}$ (and similarly for c)? Otherwise L123 is in error.*

Absolutely. We thank the referee for noting this typo which we have corrected.

*L151 – I think this is a fine assumption to make but worth discussing more. If there is in reality a vertical concentration gradient, then essentially the parameter inversion will overestimate the settling velocity by a factor equal to the ratio of near-bed to flux- averaged SSC, correct? Grams and Wilcock (2007) is relevant in discussing vertical concentration profiles in rivers with macro-roughness.*

The referee is correct. We have added a paragraph about this point at the end of section 5.1.1.

*Table 1: Could a column for grain size (inferred from $V_s$) be added?*

We have added a column to Table 1, with the grain size inferred from $V_s$.

*Table 1: should be Fig. 5, not Fig. 4.*

Thank you for noting this mistake, which has been corrected.

*L256: "In this dilute regime, sediments settle at a velocity equal to the settling velocity of a single particle" – I'm not sure that I understand this sentence. I agree with interparticle interactions being negligible, but it does not follow that sediment settling can be represented by a single settling velocity. E.g., considering the case of particles settling out of suspension without re-entrainment, the concentration of each settling-velocity class of particle declines exponentially with a rate constant proportional to the settling velocity, but the decline in total sediment concentration (i.e. all settling velocity classes summed) is not itself exponential and therefore cannot be represented with a single settling velocity.*

This sentence was indeed ambiguous. The fact that interactions between particles are negligible allows us to assume that the settling velocity is independent of the concentration. As pointed out by the referee, the suspended load usually consists of grains of different sizes, which thus settle with different velocities. Here, however, we further simplify our model, and assume that the decline in total sediment concentration can be represented with a single representative settling velocity, $V_s$. This is, again, a strong assumption, which calls for a discussion of its limits. We have therefore modified section 3 and section 5.1 to clarify these assumptions, and discuss their limits.

*L283 – typo, "not" is missing*

Thank you for noting that typo. As a matter of fact, the corresponding sentence has changed and the text is now different.

*L283 – Worth mentioning that this is not an abrupt transition: between 5 and 10 microns there is a clear decline in SSC, it's just slower.*
*&*
*L284: "which do not settle on the bed" – clarify that this is an interpretation based on the data. There is a growing body of work suggesting that this conceptualization of washload is perhaps over-simplified (e.g. Ren and Packman 2007, Dallman et al. 2020). Also, does the wash load pick up at the same threshold water level as the rest of the sediment?*

We agree with these two comments and have reworked the corresponding paragraph so as to clarify the different points raised by the reviewer. The text now explicitly states that the transition between the two behaviors is a smooth one. It also clarifies that assimilating particles of size less than $10\mu m$ to the washload is but our oversimplified interpretation of the data. Regarding the threshold water level at which the wash load picks up, our data only show that its concentration, although small, is not zero, before the flood. We have added this information in the text.

*L287-288: I think this is overstated; it shows that the inferred grain sizes are within the range observed in the field but not necessarily that the model can be used to accurately infer a characteristic size of suspended sediment. That said, the possibility of using the model to infer grain size would be an excellent discussion point.*

We have suppressed the corresponding sentence.

*A broader discussion of how the model represents a grain size distribution as a single characteristic grain size and why this seems to work would be warranted. Where does "washload" fit in – does the washload component need to be subtracted out, or does the model work well despite it? What kinds of grain size distributions might "break" the model? Does flocculation matter (Lamb et al. 2020)?*

Sections 5.1.1. and 5.3 of the new manuscript discuss explicitly all these points. In short, we expect the model to work as long as the grain size distribution is narrow. The model cannot account for the washload. It does not account either for flocculation.

*L362-363: In other words, supply limitation? This sentence would be worth expanding into a short paragraph of discussion.*

Sections 5.3 now contains a paragraph where we expand on how we could incorporate supply limitation in the model.

**2   Reviewer 2**

*General comments The authors present a new model to simulate suspended sediment concentration based on water depth time series, for floods that exhibit a counterclockwise hysteresis pattern between concentration and discharge. After calibration on the Capesterre tropical catchment, the model appears to be able to simulate correctly such hysteresis patterns, and the concentration values. Moreover, the calibrated parameter values are consistent with physical explanations of river processes. The model is also tested against another catchment where it provides consistent results. The paper addresses the issue of suspended sediment modeling, which is clearly relevant to the scope of SURF. The proposed modeling approach is novel, yield convincing results, and will certainly of great interest for the community. The paper is clearly written. My main concern is that I find this paper is not well rooted into the literature, particularly concerning suspended sediment modeling. This should be improved both for the introduction and discussion sections (see details below) in order to make the paper stronger. Moreover there are several points that deserve more accurate explanations and discussion (the concentration-turbidity relation, the model assumptions, the quantification of model performance, see line by line comments) and some conclusions that are a bit too quick. I therefore recommend major revisions before accepting this article in ESURF journal.*

We thank the reviewer for his/her substantiated comments, which helped us improve the manuscript in many ways. Her/his main comment about the need of a more thorough discussion led to substantial modifications of the manuscript. In particular, the introduction now contains a series of paragraphs that review the different approaches for modeling suspended sediment transport. as suggested by the reviewer, the new manuscript also contains a discussion section.

*Introduction: I find that the authors do not relate their work to existing literature on suspended sediment concentration or yield models. There is a large body of research concerning the modeling of suspended sediment yield, with either advection-settling-erosion equations, concentration-discharge relations, empirical models at various time-scales .... Therefore, it would be useful for the reader to provide some background on existing approaches, explain their limitations and show how the proposed approach brings novelty and improvements. I am totally convinced that it does bring novelty but this has to be explained in relation to the literature.*

We have modified the introduction by adding a series of paragraphs that review the different approaches for modeling suspended sediment transport. We hope that they provide the background necessary to clarify the novelties, the advantages and the limits of the model we propose.

*Discussion: Currently, the main discussion is about the meaning of the parameters (section 5). Even if this is a very interesting section, I think that the paper deserves a larger discussion section, including (3) a discussion of what this new model brings in comparison to existing models of suspended sediment yield, (4) a discussion on the limitations of this model, in particular concerning the validity of the assumptions, and the uncertainty related to the concentration-turbidity relation, and (5) some perspectives and applications of such model, for instance at larger time and spatial scales. I suggest adding a full discussion section to the paper, which could incorporate the existing sections 5 (meaning of parameters) and 6 (test against another data set), and address all the other points mentioned above.*

Following the reviewers advice, we have completely reorganized the manuscript. The latter now contains a large discussion (section 5) which includes the analysis of the physical meaning of the parameters (section 5.1), the test of the versatility of the model through its application to the Draix catchment (section 5.2), and a discussion on the advantages and limitations of the model (section 5.3).

We elaborate a bit on the perspectives and applications of our model both in section 5.3 and in the conclusion.

*Structure: I might be a bit conservative but I am not totally convinced by the current structure of the paper. It reads well but I find that it does not make clearly visible what the results of the paper are. Is the field data from Capesterre a result from this paper, or a preliminary summary of previous studies, only presented here as an introduction for the modeling work? I would suggest using a more standard organization with a methodological section (field setting, field data acquisition, model construction and calibration), a result section (field data sets and calibration results) and a full discussion section that could address all the points mentioned earlier. This is not a critical point for publication however and I would let the editor comment on the necessity, or not, to change this structure.*

We would rather keep the current organisation of the article, as we believe it makes for easier reading. That said, we understand the referee's concern about "what the results of the paper are". The previous version of the manuscript did not convey clearly the focus of the paper. We have therefore reworked the last paragraph of the introduction to clarify that the focus of the paper is on the model, not on an in-depth analysis of field data.

*Line by line comments*

*Line 2: "the model correctly represent the transport of suspended sediment": I think this sentence is not exact; the model actually does not represent the transport of sediment itself. I suggest replacing by "the model correctly represents the evolution of sediment concentration at the outlet"*

We have modified the sentence which now reads "The model correctly represents the concentration of suspended sediment during floods...."

*Line 6: "both of which are related to the availability of fine sediment": I don't think that this is demonstrated in the paper. I suggest removing this from the abstract.*

We have removed this sentence.

*Line 47: Perhaps worth mentioning that the distance of sediment sources to the outlet is an important factor of hysteresis. And that this effect can be coupled with the spatial heterogeneity of rainfall.*

We have modified the corresponding sentence to stress this point. It now reads : "In this case, the spatial distribution of rainfall in the catchment area and the distance between the sediment sources and the sampling point influence the shape of the concentration-discharge relationship in a complex way"...

*Line 51: Here I think that the authors should review existing approaches for modeling suspended sediment transport. There had been a lot of research on this topic so they should at least explain what the limitations of existing approaches are and why they choose to build a new type of model.*

As discussed previously, we have modified the introduction by adding a series of paragraphs that review the different approaches for modeling suspended sediment transport.

*Line 94 and figure 1d: The calibration of the turbidity-concentration relationship is quite noisy. Could the author introduce an estimation of the uncertainty for the concentration extracted from this relation?*

We added the uncertainties on the parameters.

*Moreover, is there any information available on the variability of suspended sediment grain-size during the floods? It is sometime argued that hysteretic patterns in concentration may result from changes in the turbidity-concentration relationship due to grain-size variability (Landers and Sturm, 2013, https://doi.org/10.1002/wrcr.20394). Figure 1d suggests that there is a great variability in this concentration-discharge relationship, which could be related to a variability in grain-size. This is an important point to check before attributing the hysteresis in turbidity to a hysteresis in concentration, which is the focus on the paper.*

*Update: having read the whole paper and seen figure 6, it seems that there is indeed a shift in grain-size during the flood. I suggest that the authors use this data to estimate a mean grain-size on the rising and falling limb, then search for information from the sensor manufacturer, and/or from the literature, to estimate if this grain-size shift could affect, and how much, the turbidity-concentration relation of Figure 1d.*

Absolutely this is a good point. Hence we added a discussion on LISST measurements that testify that at a flood event scale, the $d_{5}0$ of the suspended load is extremely stable.

*Line 95: Is the field data on water level and concentration already published somewhere, or could it be published with this paper? I find it a little frustrating to read about a 3 year data set with 217 floods without seeing more than what is displayed in Figure 1e, 2 and 4. Even if I trust the authors concerning their analysis on flood extraction and hysteresis calculations, I think it would be a good thing to make this data set available, both for the reader, and for the data producers themselves.*

The data used in this paper were collected by two observatories : the Observatoire de l'Eau et de l'éRosion aux Antilles (ObsERA, INSU-CNRS, `http://webobsera.ipgp.fr/`), located in Guadeloupe, and the Draix–Bléone Critical Zone Observatory (`https://bdoh.irstea.fr/DRAIX/`), located in the french Alps. These data are freely available from the website of each observatory. We have modified the acknowledgement section to clarify this point.

*Figure 2: The three floods that are selected here have a very simple shape, with an abrupt increase and an "exponential-like" decline. Are these floods representative of the whole data set, do most floods exhibit the same shape in this catchment? Are there any multipeak events?*

We perform a visual inspection over the data and detect just a few examples with multipeaks events. Hence we decided for sake of clarity to avoid these specific events in our analysis.

*Line 145-147: Do you think that this bed replenishment from hillslope really occur gradually, with no effect during floods? I would expect that hillslope processes such as landslides or debrisflows should be more frequent during strong rainfall, therefore could also represent sediment sources during floods on top of the riverbed source itself.*

We agree with the referee and have added the following sentences to stress this point : "During extreme rainfall events, landslides and debris flows may feed the river with large amounts of sediment. When

this happens, the sediment bed cannot be considered the only source of sediment, and our model likely fails to represent the concentration of suspended sediment."

*Line 148: If available, an information on suspended sediment grain-size would be useful here. If it is mostly silt, rather than sand, it would reinforce this assumption*

We have modified the paragraph which now reads : Particles in suspension in the Capesterre river mainly consist of silt and fine sand, with a median grain size of 34.8 $\mu m$ (see section 5.1). Besides, the water level rarely exceeds 1 meter, a value comparable to the median grain size of its sediment bed, of the order of ten centimeters....

*Line 152-155: This is indeed a very strong assumption, which is not realistic at all to me. It raises the issue of water mass conservation. With a uniform rain, the amount of water arriving at one point should scale with drainage area. Therefore, it is not possible for the water depth to be always the same at every point in the system. Even if you argue that the river width increases downstream, this increases is generally thought to scale with drainage area to a power ¡ 0.5 (see Lague 2014, DiBiase and Whipple, 2011 for instance) therefore this increase is not sufficient to conserve water unless water depth increases with drainage area too.*

*To my mind, this cannot be presented as an "simplifying assumption driven by field observation" (line 140) since it does not seem to be rooted in any observations, physical principles, or expected behavior of the system.*

We did not mean that the flow depth is the same at every point in the system, but, rather, that the water level rises uniformly, i.e. with the same rate $dh/dt$ everywhere. That being said, we have suppressed this entire paragraph and rewritten the derivation of the model, which now contains a scaling analysis that focuses on the magnitude of the advective term (see below).

*Perhaps an alternative approach to simplify the model would be to write the full equation, then make the assumption that the advection terms are negligible (with no a priori physical basis), then check that this is indeed the case after solving the model. This could be done by comparing estimates of advective fluxes, and erosion and deposition rates after calibration.*

It is not the advective flux of sediment that matters, but its gradient. Indeed, the advective term corresponds to the gradient of sediment flux in the streamwise direction. Evaluating its magnitude in the field would require measurements of the concentration of suspended sediment at several positions along the river course. Unfortunately, this cannot be done for the Capesterre river, which is equipped with a single station. Yet the referee's suggestion motivated us to clarify the conditions in which the advection terms are negligible. Following this comment, we have entirely rewritten the derivation of the model. As suggested by the referee, we start with the "full" equation, which includes the advective term. We then perform a scaling analysis that allows us to evaluate its magnitude relative to the other terms in the equation. We find that the magnitude of the advective term is controlled by the ratio of two characteristic times. The first one is the duration of the flood, $\tau_f$. The second one is a characteristic settling time, $\tau_s$, defined as the time necessary for a particle to settle at the representative settling velocity $V_s$ over a height equal to the threshold height, $h_t$. When the ratio of these two time scales, $\tau_s/\tau_f$, is small, as is the case in the Capesterre river, the advective term is negligible.

*Line 156: Removing the advection terms: does this imply that there is no flow velocity at all (which sounds problematic since bottom shear stress (induced from flow velocity) is responsible for sediment entrainment? Or is it that flow velocity and concentration are the same upstream and downstream therefore the advection contribution is null ? I understand that it should be the second option, but this should be made clearer to the reader.*

The advective term corresponds to the streamwise gradient of sediment flux. To neglect it is to assume that the flood wave is so long that this gradient becomes negligible compared to the entrainment and deposition rates. The new version of the model derivation contains a paragraph that clarifies this.

*Line 167: Perhaps you could detail that an increase of h implies an increase of the flow velocity hence of bottom shear stress?*

We have modified the derivation of the expression of the entrainment rate to clarify the link between flow depth and stress. In particular, we have added the following sentence : "... we note that an increase of water level often implies an increase of bottom shear stress. In the following, we shall therefore use water level as a proxy for shear stress."

*Line 168: This form of erosion rate is not new, I think it is a classical form for erosion rate as a function of a bottom shear stress in the sediment transport literature. The author could acknowledge here that they build the model based on these existing approaches.*

Absolutely. We have added the following sentence to clarify this point :" Keeping in mind that the water level h acts as a proxy for the shear stress, equation (4) is similar to expressions of the entrainment rate commonly used in the sediment transport literature (Bagnold, 1956; Van Rijn, 1986, 2007; Claudin et al., 2011)."

*Line 204 and further: Why using "assimilation" instead of "calibration"? As far as I understand, the procedure described here is a model calibration?*

We agree with the referee and we changed this terms and rephrased the associated sentences.

*Line 209-211: "the model reproduces surprisingly well" and "the model accounts reasonably well": could you quantify how well does the model perform, using a quantitative index? For instance, a RMSE, or other. This would be useful for comparison with the results over a longer period, and the results on the Laval catchment.*

We provide the adjustment fit score into the revised version.

*Line 215: Any suggestion to address this bias? Perhaps this could be discussed in a future discussion section?*

This would require to have a different sampling rate during the rise compared to the tail of the flood. Numerically, this has been done and we verified that the models still capture similarly the observed behaviour, while have the same parameters.

*Line 225: It would be interesting to compare the model results with the results from a simple rating curve approach (fitting a relation between Q and C), both at the scale of the event, and of the chronicle.*

As requested, we now compare the performance of the model with the more classical rating-curve approach, over the series of floods of Fig. 5. We discuss the results of this comparison in sections 4.3 and 5.3.

*Line 230: is it possible to run the same test over a full year of data ?*

Numerically, there is no limitation. However, our dataset suffers from numerous gaps and missing data that make such a test hardly possible. In addition, 40% of the floods produce clockwise loops of the concentration vs water level relation (section 2.2). Such events cannot be correctly modeled by our approach. A run of our model over a full year is therefore likely to fail to represent the data.

*Line 233: "reasonably well": can you quantify this?*

Again, this is visually shown in figure and we also added the score value into the revised text.

*Line 240: Would a model based on a simple rating curve perform as well, or better, over this period? The present model definitely brings some improvement in terms of simulating the hysteresis, but does it also improve the simulation of total sediment export compared to a simpler model? This could be quantified by calibrating a simple power law relation between discharge and concentration, calibrating this model against the same data and computing the same performance index (RMSE or other) than for the present model.*

As mentioned above, we now compare the performance of the model with the more classical rating-curve approach, over the series of floods of Fig. 5. We discuss the results of this comparison in sections 4.3 and 5.3.

*Line 248: "we suspect that the characteristic erosion rate and the exponent n reflect changes in the sediment availability": Could you explain a bit more? Why? Any suggestion to test this hypothesis?*

The characteristic entrainment rate $\epsilon$, the exponent $n$, and the threshold of water level, $h_t$ parameterize the expression of the rate at which particles are entrained from the bed. Any change of these parameters thus reflects a modification of the conditions in which fine sediment is entrained from the bed. In practice, we know that the entrainment rate depends on two things : the threshold of grain entrainment and the availability of fine sediment. In our model, the threshold of water level, $h_t$, accounts for the threshold of grain entrainment. This is the reason why we propose that the characteristic entrainment rate $\epsilon$ and the exponent $n$ might be related to the sediment availability. However, this proposition is purely speculative. We are afraid that, at this stage, we have no feasible suggestion to test this hypothesis. We have modified the corresponding paragraph to clarify this point.

*Line 261-270 : Here you use an expression for the drag force (equation 7) which is correct for a turbulent flow, but the particle Reynolds number (Rep = Vs ds/nu) is smaller than 1. Therefore you should rather use the classical Stokes formula, Vs = (s-1)gD2̂/(18 nu). (And I don't think that Andreotti et*

*al (2013) were the firsts to introduce Stokes' law by the way :-) )*

The referee is mistaken. Equation (7) – now equation (9) – is not the expression for the drag force in a turbulent flow. Together with the expression of the drag coefficient (8) (now equation 10), it provides a general expression of the drag coefficient, first introduced by **?**, that represents reasonably well the drag force over a large range of Reynolds numbers (from $Re = 0$ to $Re \approx 5000$). In particular, in the limit of high Reynolds numbers, the expression of the drag coefficient reduces to:

$$C_d = C_\infty \, , \tag{1}$$

so that the drag force reads

$$F_d = \frac{\pi}{8} C_\infty \, \rho d_s^2 V_s^2 \, . \tag{2}$$

This is, indeed, the expression for the drag force in a turbulent flow.

At small Reynolds number, however, the expression of the drag coefficient becomes :

$$C_d = \frac{24}{Re} \, , \tag{3}$$

so that we recover the classical Stokes formula for the drag force :

$$F_d = 3\pi\mu d_s V_s \, . \tag{4}$$

The advantage of this formulation is that it allows us to derive an expression for the settling velocity that does not require a priori knowledge of the particle size. This point is crucial as we are looking for an expression of the grain size as a function of the settling velocity. We have modified the derivation of the expression of the grain size as a function of the settling velocity to clarify these points.

*Line 290: Do you mean that the water level is a proxy of the bed shear stress?*

Indeed, this is exactly what we mean. Here, the word "but" means "no more than" or "only". To avoid any ambiguity, though, we have replaced "but" with "just" :"In the phenomenological model developed in section 3, the water-level is just a proxy for the shear stress $\tau$ that the river exerts on its bed."

*Line 297: This equation (12) requires an assumption of flow uniformity, which is worth mentioning to the reader.*

We have modified the corresponding sentence into :"The width of the Capesterre river is constant over a few hundred meters upstream of the gauging station. There, the flow is approximately uniform. This allows us to approximate the threshold shear-stress by ..."

*Line 301: What is the resolution of the DEM that is used to compute this slope? It is usually quite difficult to obtain accurate river slope values from a DEM unless the resolution is really fine. Moreoever, using equation 12 to estimate the shear stress requires that the flow is uniform at the gauging station, which is often not true (when stations are located on a weir or in a narrowing section for instance). Is it (approximately) the case at La Digue station?*

The spatial resolution of the DEM is 5 meters, which seems accurate enough to compute an estimate of the river slope. The river keeps a constant width over a few hundreds of meters upstream of the gauging station, so that the flow is approximately uniform. We have modified the corresponding paragraph to add these informations and clarify these two points.

*Line 310-317: This is an interesting point to discuss, particularly in relation to Misset el al, GRL, 2021, that showed how bed mobility was able to release fine particles and increase concentration. However, this result is in contradiction with the first assumption of the model (Line 144) according to which there was a large amount of fine sediment available in the riverbed.*

We have added a reference to the work of Misset et al, GRL, 2021, which, indeed, supports our interpretation that the entrainment of coarse bed sediment may release fine sediment from the river bed. However, we disagree with the referee about the second part of his comment. As argued in the response to the next comment, we feel that there is no contradiction between the need to entrain coarse sediment before releasing fine particles, and the assumption of a large amount of fine sediment available in the river bed.

*I understand that for a large flood, coarse bed sediment can be entrained therefore release fine sediment from the subsurface. But there should be a range of intermediate floods for which the coarse particles do not move. For such floods, either there is a large amount of fine sediment on the bed hence the critical shear stress should correspond to a small sediment size, either fine sediments are mostly stuck in the subsurface hence the transport-limited assumption does not hold. This should be discussed into more detail. Perhaps you could calibrate the model on floods of various amplitude and check if there is a different trend in the calibration for intermediate and high floods? Otherwise, I think it is a bit too early to conclude in a general way from these 4 values of ht that "the threshold of suspended sediment transport is set by that of the coarse particles"*

The exact value of the threshold water level is indeed computed for only four floods. Yet, it is, by far, the most robust result of our manuscript: direct observation of the data show that the concentration of suspended sediment rises only once the water level exceeds a value of about 20cm, independently of the magnitude of the flood or the time of the year (Fig. 1 e and f). We have no indication of a difference of threshold between intermediate and high floods.

The referee is right : were fine sediments stored in abundance at the bed surface, one would expect the existence of two thresholds : the first one would correspond to the entrainment of fine sediment from the bed and the second one would describe the additional release of fine particles trapped below coarse sediment. Here, however, all evidence points to a single threshold.

As a matter of fact, direct observation shows that the bed of the Capesterre river consists of gravel and pebbles, with tiny patches of fine particles trapped in the space between neighboring grains. Fine particles are in direct contact with the flow. Yet, the bed roughness is so high (the median grain size of coarse particles is of the order of 10 cm) that fine sediments, of size a few hundred microns at the most, are effectively screened from the flow, and remain trapped between coarse particles until the latter start moving.

This, however, does not contradict the transport-limited assumption. Stricto-sensu, the transport-limited assumption does not necessarily imply the existence of a huge amount of fine sediment in the riverbed. Rather, it requires that the stock of available sediment is large enough not to affect sediment transport during the flood. In such a configuration, the model does not need to explicitly account for a decrease of this stock, at the scale of a flood. The fact that our model successfully accounts for counter-clockwise floods suggests that this condition is met in this case. Conversely, this is also why

we suspect – but have not proved – that clockwise hysteretic loops might result from a depletion of the stock of available fine sediment, during the flood.

We have added a few sentences at the end of section 5.1.2 to clarify these points.

*Line 399: "the model represents rather well": What do you mean by "rather well"? Could you quantify? (see earlier comment)*

Some comments as previously.

*Line 341: "the best-fit parameter values fall within the same ranges as those obtained in Capesterre". So what does it mean? Could you discuss on the potential implications of this observation? Does a similar range of falling velocity suggests that grain-sizes are the same? Does a similar range of threshold water level indicate that suspended sediment transport is also controlled by armoring in the Laval catchment?*

In all honesty, we had selected these catchments because they shared similar properties, in terms of scale, slopes and flow regime. Our results confirmed those similarities.

*Line 347: "that accounts for the transport of suspended sediment": I don't think that the model accounts for the transport (no advective terms). I suggest replacing by "that represents the temporal evolution of the concentration during floods.*

*Line 349: same comment as above*

We disagree with the reviewer. The advective term is not the sediment flux itself, $F$, but its derivative in the streamwise direction $dF/dx$. Therefore, our model does not assume that the sediment flux is negligible – a poor assumption indeed. It only assumes that the gradient of flux is small. As noted by the referee, our model is capable of representing the temporal evolution of the concentration during floods. Once we have this concentration, it suffices to multiply it by the river discharge to get the sediment flux. As a result, our model does account for the transport of suspended sediment, as long as the assumptions that support it are met.

*Line 356: This has only been suggested in line 248 but has not been discussed or demonstrated. I don't think that this should be part of the conclusions. "We suspect that" without any other kind of explanation is not a scientific result in my opinion.*

We agree with the referee that this assertion has not be demonstrated. This is why we use the word "suspect". To emphasize that this a mere speculation, we have modified the end of the conclusion which now reads "As for the entrainment rate $\epsilon$ and the exponent $n$, we suspect that they are linked to the availability of fine sediment in the river bed. If these conjectures prove to be correct, our model might offer the opportunity to detect river-wide changes, like, for example, modifications of the sediment size and availability induced by bank incision or landsliding."

*Line 365-368: I think these points deserve to be addressed in a discussion section. In particular, a discussion concerning the limitations of the model, and the effects of grain-size, should be developed in the paper earlier than in the conclusions.*

Following this comment, we now discuss these points in more details in section 5.3 of the discussion section.

---

## Referee Report (RR1)

I have carefully reread the new manuscript and the authors' response to reviewers. My original comments about the scientific value of this work stand, and I am pleased to see that the authors have greatly improved and expanded the introduction and discussion to better contextualize this work and address salient questions. I have a few minor comments but I don't think another round of review is necessary; this is a valuable work worthy of publication.

Line numbers refer to the tracked changes version.

L92: "before concluding" – you could tease in short here what the most important conclusions of this study are.

L134: is this a way of notating the uncertainty in that measurement? Maybe add + to the superscript and – to the subscript to make that more clear.

L145, fig 2: I still think a dashed vertical line in the left subpanels aligned with the flood peak would help to visualize the time lag of the sediment peak.

[Figure]

L153: "ignoring the washload" – can this be justified in some way? How much does the washload concentration change much during the flood? My concern is that turbidity is often extra sensitive to the finest sediment.

Fig 3: I know it's the same as fig 2 but nevertheless the figure should have a legend.

Fig 2, Fig 3: Add null model (c=f(h)) as you did with fig 5?

L173-174, Fig 4: consider adding as a subpanel a cartoon of how the hysteresis index works. For me equation 1 was not obvious, but a visual example of the integral makes it immediately obvious why it works.

L257-276 very clear explanation!

L279: suggest omitting "bed" from "bed area" as this makes me think you're talking about concentration in the bed rather than in the flow

L308, L311: I think "calibration" would be a better word than "adjustment"

L446: "there, the flow is approximately uniform" – what is this based on?

L505: focusing

L530: why would the simple rating curve systematically overestimate? This seems like a problem with the fit; I would think some floods would be overestimated and some underestimated, with higher RMSE than your method, but not necessarily a systematic difference.

---

## Referee Report (RR2)

**Report on the revised version of article "Phenomenological model of suspended sediment transport in a small tropical catchment" by Amande Roque-Bernard et al.**

First, I congratulate the authors for the improvements made to this revised version and particularly concerning the introduction and discussion revisions. I also thank the authors for the detailed and convincing answers to my questions.

I still have a few small remaining comments:

Line 153: Why should you ignore the washload considering when considering the possible influence of grain-size on the concentration/turbidity relation?

Line 324: I am not familiar with this chi2 measure of the quality of the fit. Could you clarify what it means when you say that chi2 < 0.04 for the top 25% of you adjustments? Does it mean that the remaining 75% of the adjustments are not significantly different from a random model?

Line 535: I am surprised to read that the model provides a better estimation of the sediment flux than a rating curve for cases with clockwise hysteresis. How was this rating curve calibrated? Was it based on turbidity and discharge measurements for the event only, or over the full year? Could this rating curve be biased by the fact there are more measurement points in the falling than in the rising part of the hysteresis?

---

## Author Response (AR2)

**1 Responses to reviewer 1**

I have carefully reread the new manuscript and the authors' response to reviewers. My original comments about the scientific value of this work stand, and I am pleased to see that the authors have greatly improved and expanded the introduction and discussion to better contextualize this work and address salient questions. I have a few minor comments but I don't think another round of review is necessary; this is a valuable work worthy of publication.

We thank you for your appreciation of the revised manuscript. We detail below how we addressed your additional comments.

*L92: "before concluding" – you could tease in short here what the most important conclusions of this study are.*

We have modified part of the last paragraph of the introduction to tease the most robust conclusions of our study.

*L134: is this a way of notating the uncertainty in that measurement? Maybe add + to the superscript and – to the subscript to make that more clear.*

This notation is not a way of notating uncertainties, but a way of indicating a range of variations, sometimes used in planetology. We understand that this is confusing. Accordingly, we have rephrased the text which now reads "It is virtually zero 98% of the time, and rises only during floods, where it may reach up to 1430 mg $L^{-1}$."

*L145, fig 2: I still think a dashed vertical line in the left subpanels aligned with the flood peak would help to visualize the time lag of the sediment peak.*

This is done for fig. 2 and Fig. 3, thank you for the suggestion.

*L153: "ignoring the washload" – can this be justified in some way? How much does the washload concentration change much during the flood? My concern is that turbidity is often extra sensitive to the finest sediment.*

We agree and have modified the text accordingly. In fact, the data at our disposal show that the distribution of sediment size is bimodal, with two peaks. The first one, around 5 micrometers, corresponds to the washload, and remains remarkably stable during the flood. The position of the second peak varies between 23 and 32 micrometer for the data at our disposal. Assuming that these data are representative, we expect no significant change of the turbidity-concentration relationship during a flood. We have modified the text to clarify this point.

*Fig 3: I know it's the same as fig 2 but nevertheless the figure should have a legend. Fig 2, Fig 3: Add null model (c=f(h)) as you did with fig 5?*

We have added a legend on Figure 3, as requested. However, the addition of the null model to Figures 2 and 3 makes these figures cumbersome, without providing any new information : these figures display loops that cannot be represented by the null model. For this reason, we prefer to restrict the discussion of the null model to the time series of fig. 5, only.

*L173-174, Fig 4: consider adding as a subpanel a cartoon of how the hysteresis index works. For me equation 1 was not obvious, but a visual example of the integral makes it immediately obvious why it works.*

We added a subpanel on both sides of the $H_I$ graphically explaining the calculation.

*L257-276 very clear explanation!*

Thank you for your appreciation.

*L279: suggest omitting "bed" from "bed area" as this makes me think you're talking about concentration in the bed rather than in the flow.*

Done.

*L308, L311: I think "calibration" would be a better word than "adjustment"*

We have rephrased this sentence which now reads "In the next section, we discuss the calibration of our model on field data, over a short period of time corresponding to a single flood event.

*L446: "there, the flow is approximately uniform" – what is this based on?*
It is an assumption. We have clarified this point by rephrasing the sentence which now reads : "There, assuming that the flow is uniform, we approximate the threshold shear-stress by ...

*L505: focusing*
Corrected.

*L530: why would the simple rating curve systematically overestimate? This seems like a problem with the fit; I would think some floods would be overestimated and some underestimated, with higher RMSE than your method, but not necessarily a systematic difference.*
We constructed the rating curve from direct measurements of the concentration of suspended sediment in water samples collected by an automatic sampler all through the year. The automatic sampler is equipped with a pressure probe, which triggers the collection of 24 water samples, one every 15 to 60 minutes, as soon as the rivers stage exceeds a threshold set by the operator. In the dataset at our disposal, this threshold varies between 30 and 80 cm, that is way above the threshold of entrainment in the Capesterre river. Because of this, most measurements belong to the falling rather than to the rising part of the hysteresis. This biais likely explains why the rating curve overestimates the concentration of suspended sediment in the river. We have modified the corresponding paragraph to clarify this point.

**2   Responses to reviewer 2**

*First, I congratulate the authors for the improvements made to this revised version and particularly concerning the introduction and discussion revisions. I also thank the authors for the detailed and convincing answers to my questions.*
We thank you for your appreciation of the revised manuscript. We detail below how we addressed your additional comments.

*Line 153: Why should you ignore the washload when considering the possible influence of grain-size on the concentration/turbidity relation?*
You are right. We have modified the text accordingly. In fact, the data at our disposal show that the distribution of sediment size is bimodal, with two peaks. The first one, around 5 micrometers, corresponds to the washload, and remains remarkably stable during the flood. The position of the second peak varies between 23 and 32 micrometers. Assuming that these data are representative, we expect no significant change of the turbidity-concentration relationship during a flood. We have modified the text to clarify this point. Given the data at our disposal, we cannot investigate in details the evolution of the grain size distribution during floods. In this context, the fact that direct manual measurements of the concentration of suspended sediment confirm the existence of hysteretic loops remains our best argument against the possibility of an artifact.

*Line 324: I am not familiar with this chi2 measure of the quality of the fit. Could you clarify what it means when you say that chi2 ¡ 0.04 for the top 25% of your adjustments? Does it mean that the remaining 75% of the adjustments are not significantly different from a random model?*
We acknowledge that this sentence is misleading. It is actually inherited from a preliminary investigation of the methods of inversion of our model. The bottom line is that the adjustment procedure works well, and that the model correctly fits the observation. So, for the sake of clarity, we removed this sentence and refer to the summary table of our results.

*Line 535: I am surprised to read that the model provides a better estimation of the sediment flux than a rating curve for cases with clockwise hysteresis. How was this rating curve calibrated? Was it based on turbidity and discharge measurements for the event only, or over the full year? Could this rating curve be biased by the fact there are more measurement points in the falling than in the rising part of the hysteresis?*

We constructed the rating curve from direct measurements of the concentration of suspended sediment in water samples collected by an automatic sampler all through the year. The automatic sampler is equipped with a pressure probe, which triggers the collection of 24 water samples, one every 15 to 60 minutes, as soon as the rivers stage exceeds a threshold set by the operator. In the dataset at our disposal, this threshold varies between 30 and 80 cm, that is way above the threshold of entrainment in the Capesterre river. Because of this, most measurements indeed belong to the falling rather than to the rising part of the hysteresis, as you suggest. This biais likely explains why the rating curve overestimates the concentration of suspended sediment in the river. We have modified the corresponding paragraph to clarify this point.